# The methyltransferase METTL9 mediates pervasive 1-methylhistidine modification in mammalian proteomes

Erna Davydova [1,13], Tadahiro Shimazu [2,13], Maren Kirstin Schuhmacher [3], Magnus E. Jakobsson [4,5], Hanneke L. D. M. Willemen [6], Tongri Liu[7], Anders Moen[1], Angela Y. Y. Ho[1], Jędrzej Małecki [1], Lisa Schroer [1], Rita Pinto[1,12], Takehiro Suzuki[8], Ida A. Grønsberg[1], Yoshihiro Sohtome [9,10], Mai Akakabe[9], Sara Weirich[3], Masaki Kikuchi [11], Jesper V. Olsen [4], Naoshi Dohmae[8], Takashi Umehara [11], Mikiko Sodeoka [9,10], Valentina Siino[5], Michael A. McDonough [7], Niels Eijkelkamp [6], Christopher J. Schofield [7], Albert Jeltsch [3,14 ✉], Yoichi Shinkai [2,14 ✉] & Pål Ø. Falnes [1,14 ✉]

Post-translational methylation plays a crucial role in regulating and optimizing protein function. Protein histidine methylation, occurring as the two isomers 1- and 3-methylhistidine (1MH and 3MH), was first reported five decades ago, but remains largely unexplored. Here we report that METTL9 is a broad-specificity methyltransferase that mediates the formation of the majority of 1MH present in mouse and human proteomes. METTL9-catalyzed methylation requires a His-x-His (HxH) motif, where "x" is preferably a small amino acid, allowing METTL9 to methylate a number of HxH-containing proteins, including the immunomodulatory protein S100A9 and the NDUFB3 subunit of mitochondrial respiratory Complex I. Notably, METTL9-mediated methylation enhances respiration via Complex I, and the presence of 1MH in an HxH-containing peptide reduced its zinc binding affinity. Our results establish METTL9-mediated 1MH as a pervasive protein modification, thus setting the stage for further functional studies on protein histidine methylation.

[1] Department of Biosciences, Faculty of Mathematics and Natural Sciences, University of Oslo, 0316 Oslo, Norway. [2] Cellular Memory Laboratory, RIKEN Cluster for Pioneering Research, Wako, Saitama, Japan. [3] Department of Biochemistry, Institute of Biochemistry and Technical Biochemistry, University of Stuttgart, Allmandring 31, 70569 Stuttgart, Germany. [4] Proteomics Program, Faculty of Health and Medical Sciences, Novo Nordisk Foundation Center for Protein Research (NNF-CPR), University of Copenhagen, Blegdamsvej 3B, 2200 Copenhagen, Denmark. [5] Department of Immunotechnology, Lund University, Medicon Village, 22100 Lund, Sweden. [6] Center for Translational Immunology (CTI), University Medical Center Utrecht, Utrecht University, 3584 Utrecht, EA, The Netherlands. [7] Chemistry Research Laboratory, Department of Chemistry, University of Oxford, Oxford, UK. [8] Biomolecular Characterization Unit, Technology Platform Division, RIKEN Center for Sustainable Resource Science, Wako, Saitama, Japan. [9] Synthetic Organic Chemistry Laboratory, RIKEN Cluster for Pioneering Research, Wako, Saitama, Japan. [10] RIKEN Center for Sustainable Resource Science, Wako, Saitama, Japan. [11] Laboratory for Epigenetics Drug Discovery, RIKEN Center for Biosystems Dynamics Research, Yokohama, Japan. [12] Present address: Department of Molecular Oncology, Institute for Cancer Research, Oslo University Hospital, Oslo, Norway. [13] These authors contributed equally: Erna Davydova, Tadahiro Shimazu. [14] These authors jointly supervised this work: Albert Jeltsch, Yoichi Shinkai, Pål Ø. Falnes. ✉email: albert.jeltsch@ibtb.uni-stuttgart.de; yshinkai@riken.jp; pal.falnes@ibv.uio.no

Proteins are frequently modified by post-translational methylation, which is primarily found on lysines and arginines, but can also occur at other residues, such as glutamine and histidine[1–4]. The functional significance of protein methylation has been most intensively studied for lysine methylation of histone proteins, playing important roles in the regulation of gene expression and chromatin state[5]. In humans, protein methylation is mediated by a number of $S$-adenosylmethionine (AdoMet)-dependent methyltransferases (MTases), belonging to two distinct classes, the SET-domain and the seven-β-strand (7BS) MTases[3,6–9]. The SET-domain MTases mainly encompass lysine-specific histone MTases, whereas the 7BS MTases, collectively, target a wide range of substrates[6,8,9].

Histidine can be methylated at either the N1 or N3 position of its imidazole ring, yielding the isomers 1-methylhistidine (1MH; also referred to as π-methylhistidine) or 3-methylhistidine (3MH; τ-methylhistidine), respectively[1]. Histidine methylation was first described five decades ago, when it was shown that actin and myosin from muscle contain 3MH[1,10,11]. Since then, histidine methylation has been firmly established for only a few additional mammalian proteins. These are myosin light chain kinase (MYLK2) from rabbit skeletal muscle and the S100A9 subunit of the dimeric antimicrobial and immunomodulatory protein calprotectin from mouse spleen[12,13], both of which carry a 1MH-modified histidine. In addition, the B12 subunit of mitochondrial respiratory Complex I (NDUFB3) has been shown to contain multiple methylated histidines close to its N-terminus, but the exact chemical nature of the modification (1MH or 3MH) was not investigated[14]. Moreover, recent high-throughput proteomics analyses have indicated the presence of several hundred histidine-methylated proteins in humans, implying that this modification is widespread[15,16]. Importantly, studies on mammalian histidine methylation and its biological significance have so far been hampered by the absence of knowledge about the responsible MTases. However, it was recently shown that the SET-domain MTase SETD3 is the long-sought enzyme responsible for the introduction of 3MH at His-73 in actin, and that SETD3-mediated methylation plays an important role in regulating actin function and muscle contractility[16,17].

Histidine is arguably the most versatile of the proteinogenic amino acids. Both of the nitrogen atoms in its imidazole ring may become protonated, yielding neutral, and positively charged forms of histidine at physiological pH. Histidine is therefore a common catalytic residue, acting as either a general base or acid. It can also be a sensor of local pH, and is involved in a variety of other interactions, frequently including coordination of metal cations[18,19]. Against this background, histidine methylation has the potential to regulate a wide range of molecular interactions and cellular processes.

In the present work, we investigate the function of the previously uncharacterized mammalian 7BS MTase METTL9. We find that METTL9 introduces 1MH at His-x-His (HxH; where x is optimally a small residue) motifs in several different proteins from human and mouse. Amino acid analysis of total protein hydrolysates shows that 1MH is a relatively abundant modification and that METTL9 introduces the majority of 1MH in human and mouse proteomes. Moreover, the mitochondrial Complex I protein NDUFB3 is found to be an in vivo target of METTL9, and, in accordance with this, the enzymatic activity of METTL9 promotes Complex I-mediated respiration. Finally, the presence of 1MH modifications in an HxH-containing peptide is found to diminish its affinity towards zinc.

## Results

**METTL9 is a protein histidine MTase.** As part of efforts to investigate uncharacterized human 7BS MTases, we set out to study METTL9. Beyond containing the classical 7BS MTase hallmark motifs, METTL9 shows no significant sequence homology to other human 7BS MTases. However, putative METTL9 orthologues are widespread throughout eukaryotes, though notably absent from fungi and land plants (Supplementary Fig. 1).

To determine whether METTL9 is a protein MTase, we investigated the ability of a recombinant fusion protein between human METTL9 and glutathione-S-transferase (GST-hMETTL9) to methylate proteins in fractionated human cell extracts in the presence of [$^3$H]AdoMet, and detected protein methylation by fluorography. Both $METTL9$ knockout (KO) and wild-type (WT) cells were used, and an MTase inactive mutant (E174A; Supplementary Fig. 1) enzyme was included as a negative control. WT GST-hMETTL9 methylated several distinct proteins in the different cellular fractions (Fig. 1a and Supplementary Fig. 2a). This was particularly striking for the membrane fraction, where several proteins were specifically methylated by the WT enzyme and showed a stronger signal in the KO extracts (Fig. 1a), indicating that these proteins represent *bona fide* cellular substrates of METTL9. In agreement with this, a fusion protein between METTL9 and green fluorescent protein (hMETTL9-GFP) co-localized with markers for membranous compartments such as the endoplasmic reticulum and mitochondria in HeLa cells (Supplementary Fig. 2b). We also observed similar yet distinct hMETTL9-dependent methylation patterns in extracts from various tissues from WT and *Mettl9* KO mice (Supplementary Fig. 2c). These results demonstrate that METTL9 is a protein MTase responsible for methylating a multitude of substrates from different tissues and subcellular compartments.

Attempts to identify the METTL9 substrates observed in fluorography using fractionation and protein mass spectrometry (MS) were unsuccessful. Thus, we searched for substrates among previously published METTL9 interactants[20–22] by incubating several of the corresponding recombinant His$_6$-tagged proteins with recombinant His$_6$-hMETTL9 in the presence of AdoMet. In this way, we found that the poorly characterized Armadillo repeat-containing protein 6 (ARMC6) was an in vitro substrate of METTL9, and MS analysis showed that a peptide corresponding to residues 248–267 of ARMC6 became partially methylated after treatment with the MTase (Fig. 1b, Supplementary Fig. 3), with the modification either at His-263 or Asn-264 (Supplementary Fig. 4a). To pinpoint the methylation site, we introduced alanine substitution mutations at His-263, Asn-264 and surrounding residues, and tested the ability of the resulting mutant proteins to undergo GST-hMETTL9-mediated methylation using [$^3$H]AdoMet and fluorography. Wild-type ARMC6 and most of the mutants were efficiently methylated, but replacement of either His-261 or His-263 abolished METTL9-mediated methylation (Fig. 1c). Through MS analysis of tryptic peptides derived from proteins co-immunoprecipitated with hMETTL9-GFP inducibly expressed in HEK-293-derived cells, we found that ARMC6 was methylated at His-263 (Supplementary Fig. 4b), whereas the corresponding unmethylated peptide was undetectable. Thus, this shows cellular methylation of ARMC6 within the METTL9 target sequence. These results strongly indicate that METTL9 is a histidine MTase that methylates ARMC6 both in vitro and in cells.

**METTL9 methylates HxH motifs.** The methylation site of ARMC6 contains three alternating histidines (HxHxH) and, remarkably, such a pattern is also present in two of the few proteins previously reported to contain methylhistidine, mouse S100A9 and bovine NDUFB3 (Fig. 1d)[13,14]. Moreover, when searching the human proteome for instances of HxHxH, we

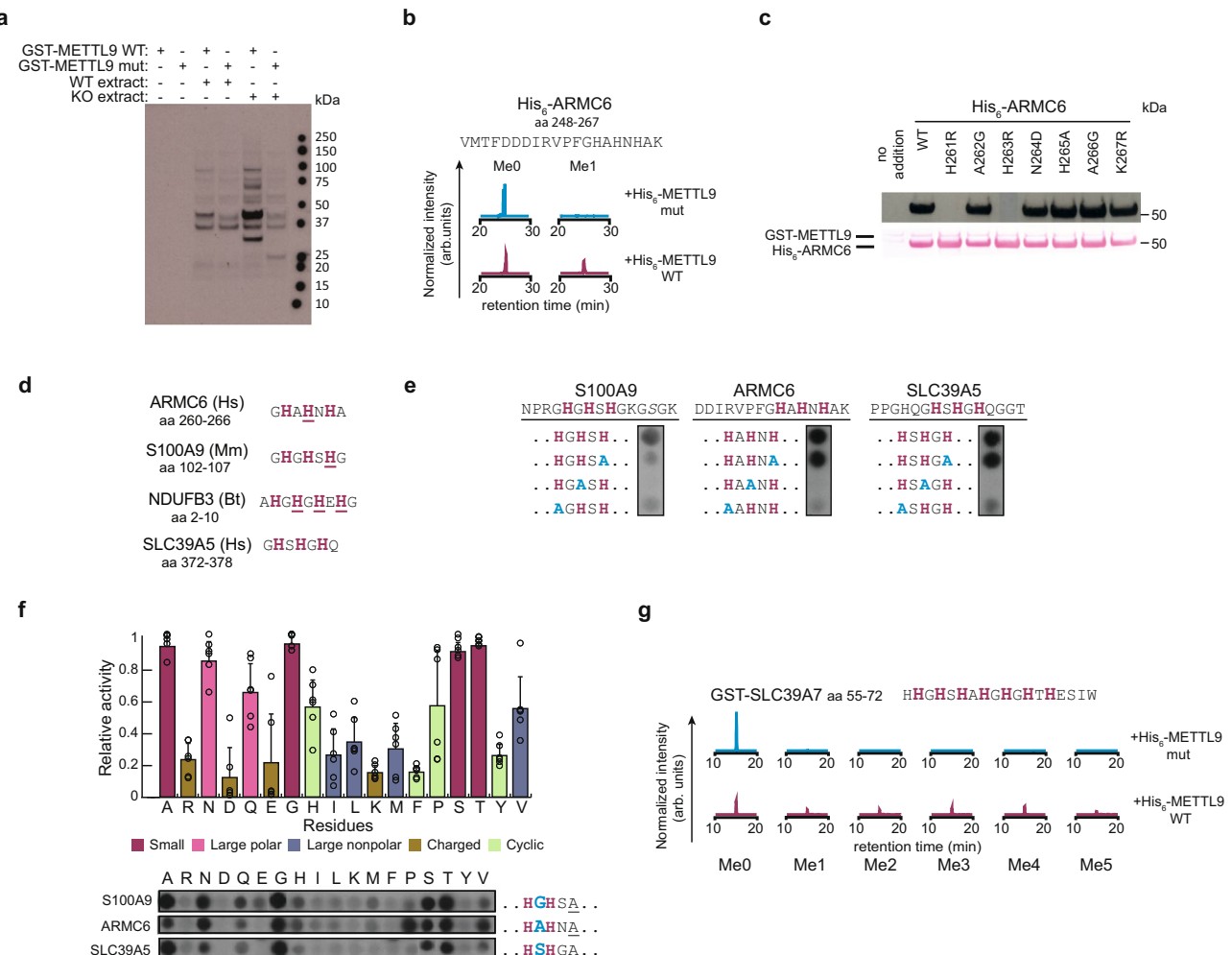

**Fig. 1 METTL9 is a protein histidine methyltransferase that methylates the ANGST HxH motif. a** Fluorography showing MTase activity, using [³H] AdoMet, of recombinant wild-type (WT) GST-hMETTL9 on proteins in the membrane fraction from WT and *METTL9* KO HAP1 cells (METTL9 mut: inactive mutant E147A). The data are representative of three independent experiments. **b** Enzymatically active METTL9 methylates ARMC6. Recombinant His₆-ARMC6 was treated with His₆-hMETTL9 in the presence of AdoMet in vitro, then digested with trypsin, and the resulting peptides analyzed by LC-MS. Normalized extracted ion chromatograms (XICs) corresponding to the monomethylated (Me1) and unmethylated (Me0) forms of the peptide ARMC6₂₄₈₋₂₆₇ are shown (full XICs are shown in Supplementary Fig. 3). **c** Fluorography showing in vitro activity of GST-METTL9 on WT and mutated recombinant His₆-ARMC6 (top). Ponceau S-stained membrane as loading control (bottom). The data are representative of three independent experiments. **d** HxHxH-containing sequences from human (Hs) ARMC6 and SLC39A5, bovine (Bt) NDUFB3, and mouse (Mm) S100A9 proteins with known methylated His underlined. **e** Activity of His₆-hMETTL9, using [³H]AdoMet, on a peptide array containing WT and His-to-Ala substituted (blue) peptides derived from S100A9, ARMC6 and SLC39A5. Similar results were observed in two independent experiments. **f** Activity of METTL9 on peptides (from **e**) modified (by His-to-Ala mutation; underlined) to contain a single HxH, and where "x" (blue, enlarged) was varied. Mean ± s.d (*n* = 6; i.e. 2 independent experiments for each of the three different peptides). Top, bar graph colored by residue type – small (A, G, S, T; purple), large polar (N, Q; pink), large nonpolar (I, L, M, V; blue), charged (R, D, E, K; brown) and cyclic (H, F, P, Y; green). Bottom, image of one of the two peptide arrays used to generate the bar graph. Note that the original image, which can be found in Supplementary Fig. 5a, has been rearranged so that each amino acid represents a distinct column. **g** Multiple methylation of an SLC39A7-derived fragment by METTL9. GST-SLC39A7₃₁₋₁₃₇ was incubated with recombinant WT or mutant His₆-hMETTL9 in the presence of AdoMet, and digested with chymotrypsin. Normalized LC-MS XICs corresponding to different methylation states of the peptide SLC39A7₅₅₋₇₂ are shown (full XICs are found in Supplementary Fig. 6). Source data are provided as a Source Data file.

found that such motifs were particularly abundant in a number of zinc transporters (SLC39A and SLC30A proteins), and notably, several of these proteins had previously been reported as METTL9 interactants[20–22]. We reasoned that HxHxH could be a recognition sequence for METTL9, and that the recombinant enzyme may be active on peptides containing this motif. To investigate this, peptides were synthesized on an array, followed by incubation with [³H]AdoMet and METTL9, and methylation was detected by fluorography. For these experiments, we used hexahistidine-tagged METTL9 (His₆-hMETTL9) which showed

considerably higher activity than GST-hMETTL9, but was unsuitable for fluorography experiments due to automethylation of the tag. Indeed, we found that His₆-hMETTL9 was able to methylate peptides derived from ARMC6, S100A9, and the zinc transporter SLC39A5 (Fig. 1e) (but no activity was observed on an NDUFB3-derived peptide). To further explore the sequence requirements for METTL9-mediated methylation, we analyzed various substituted peptides. Alanine substitutions of single histidines in the HxHxH motifs revealed that METTL9-mediated methylation was abolished on complete removal of both HxH

motifs via substitution of the middle histidine (HxAxH) (Fig. 1e). Methylation was observed for all the peptides containing a single intact HxH (AxHxH or HxHxA), albeit in some cases at a reduced level compared with the corresponding HxHxH peptide. This indicates that a single HxH is the minimal sequence motif for METTL9-mediated methylation.

Based on the above results, we set out to investigate how the identity of the middle residue, x, as well as the N- and C-flanking residues, denoted $x_N$ and $x_C$, influence methylation of the METTL9 recognition sequence ($x_N HxHx_C$). We investigated all three test sequences from above (Fig. 1e), modified to contain only a single instance of HxH (Fig. 1f, Supplementary Fig. 5a), and observed some clear trends. For the $x_N$ and $x_C$ positions, most amino acid substitutions were tolerated, but Pro and Val largely abolished methylation, and Ile or Glu at the $x_N$ position strongly reduced methylation (Supplementary Fig. 5b, c). For the middle position, x, replacement with aromatic or charged residues had a drastic negative effect on methylation, whereas the highest activity was observed with the small and uncharged residues Ala, Gly, Thr, Ser, and Asn (Fig. 1f, Supplementary Fig. 5a), very much in agreement with the presence of such residues in the identified METTL9 substrates. Based on this sequence preference (A,N,G,S,T), we refer, in the following, to histidines alternating with such residues as ANGST HxH motifs.

NDUFB3 was reported to contain three methylated histidines within the sequence HGHGHEH (Fig. 1d)[14], and we therefore set out to explore the ability of METTL9 to introduce multiple methylations in vitro. As we were unable to detect METTL9-mediated methylation of a corresponding NDUFB3-derived peptide in vitro, we instead used as substrate a sequence derived from a histidine-rich loop (amino acids 31–137) of the zinc transporter SLC39A7, since it contains a high number of ANGST HxH motifs. This sequence (as a recombinant GST fusion protein, GST-SLC39A7$_{31–137}$) was incubated with recombinant His$_6$-hMETTL9 and AdoMet, then digested with chymotrypsin, and the methylation status of the resulting peptides analyzed by MS. In general, we found the MS analysis of histidine-rich METTL9 substrates challenging, but for one of the tryptic fragments (amino acids 55–72) which contained seven alternating histidines forming six ANGST HxH motifs, extensive methylation was detected. We observed up to five methylations, with an average of ~3 methylations (Fig. 1g, Supplementary Fig. 6), as well as a considerable pool of unmethylated substrate, possibly resulting from suboptimal activity of the recombinant enzyme produced in E. coli. The above results clearly establish the ANGST HxH motif as a target sequence for METTL9, and demonstrate that METTL9 can introduce multiple methylations in stretches of alternating histidines.

**METTL9 catalyzes 1MH formation in vitro and in vivo**. Methylation of histidine can occur on the proximal or the distal nitrogen of its imidazole ring, leading to the formation of 1MH or 3MH, respectively (Fig. 2a). Thus, we set out to determine which isomer is generated by METTL9. We methylated three different HxHxH-containing peptides with recombinant His$_6$-hMETTL9 in vitro, followed by acid hydrolysis and analysis of the resulting amino acids by liquid chromatography coupled to MS (LC-MS). Indeed, 1MH was formed in all three peptides, and no 3MH was detected (Fig. 2b and Supplementary Fig. 7), in agreement with the reported presence of 1MH in the mouse S100A9 protein[13].

We reasoned that METTL9, being a broad-specificity histidine MTase, could be responsible for a substantial portion of the 1MH in the human proteome. Thus, we examined whether the total content of 1MH in cellular proteins was affected by the disruption of the METTL9 gene. We collected protein extracts from WT and METTL9 KO human HAP1 and HEK293T cells, as well as from various tissues and embryonic fibroblasts derived from WT and Mettl9 KO mice, and analyzed the amount of methylhistidine, relative to total histidine, by LC-MS/MS. Interestingly, whereas 3MH was unaffected, 1MH was reduced by ~50% in the human METTL9 KO cells, and showed an even more notable reduction in the Mettl9 KO mouse fibroblasts (Fig. 2c, d). When WT METTL9 was reintroduced into the KO HAP1 and HEK293T cells, the 1MH content was restored back up to, or even above, the WT levels (Fig. 2c). Importantly, no such effect was observed with enzymatically inactive METTL9, demonstrating that the enzymatic activity of METTL9 is required for 1MH formation in vivo. We also found a remarkable reduction of proteinaceous 1MH, but not 3MH, in various tissues from Mettl9 KO mice (relative to WT), although the amount of METTL9-dependent 1MH varied considerably between the tissues (Fig. 2d). Taken together, this indicates that METTL9-mediated methylation occurs in a wide range of mammalian tissues, and that METTL9 is the major enzyme generating 1MH in mouse and human proteins.

**Defining a catalog of METTL9 targets**. To identify additional METTL9 substrates, we employed a proteomics approach involving the AdoMet analogue propargylic Se-adenosyl-L-seleno-methionine (ProSeAM), which allows MTase-mediated transfer to relevant substrates of an alkyne moiety that can be conjugated to biotin by click chemistry[23]. Thus, we incubated protein extracts from Mettl9 KO mouse embryonic fibroblasts (MEFs) with mouse METTL9 (mMETTL9) in the presence of ProSeAM, and pulled down biotinylated proteins with streptavidin (Fig. 3a). To allow specific identification of METTL9 targets, we used a SILAC approach where proteins pulled down from a METTL9-treated heavy isotope-labeled extract were compared with those pulled down from an untreated light isotope-labeled extract (Fig. 3a). Two replicate experiments were performed, and the eight proteins identified in Experiment 1 represented a subset of the 16 proteins identified in Experiment 2. (Fig. 3b). Intriguingly, while an ANGST HxH motif is found in only ~11% of all mouse proteins (number obtained by using ScanProsite[24]), the eight proteins that were identified in both experiments all contain such a motif, and this was also the case for two of the eight additional proteins identified in Experiment 2 only. Reassuringly, NDUFB3 was among the eight proteins identified in both replicates, and so were zinc transporters from the SLC30A and SLC39A families. Taken together, these experiments identified new candidate METTL9 substrates, and further corroborated METTL9 as a broad-specificity enzyme with a preference for ANGST HxH motifs.

To further investigate candidate METTL9 substrates, corresponding peptide sequences were spotted on an array and His$_6$-hMETTL9-mediated methylation assessed using [$^3$H]AdoMet and fluorography. 56 peptides were investigated, representing (ANGST and non-ANGST) HxH-containing sequences from the following categories of proteins: (1) hits from the ProSeAM experiments; (2) METTL9 interactants[20–22]; (3) other proteins, such as reported and putative His-methylated proteins, and proteins of particular biological interest (Fig. 3c, Supplementary Table 1). As negative controls were included corresponding peptides with HxH motifs disrupted by His-to-Ala replacements. About half of the peptides were methylated, to varying extents, by METTL9, and the control peptides were not methylated in any case (Fig. 3d, e, Supplementary Fig. 8). The five peptides that showed the strongest activity stem from proteins involved in a wide range of cellular processes, i.e. chaperone DNAJB12, unconventional myosin MYO18A, two zinc transporters

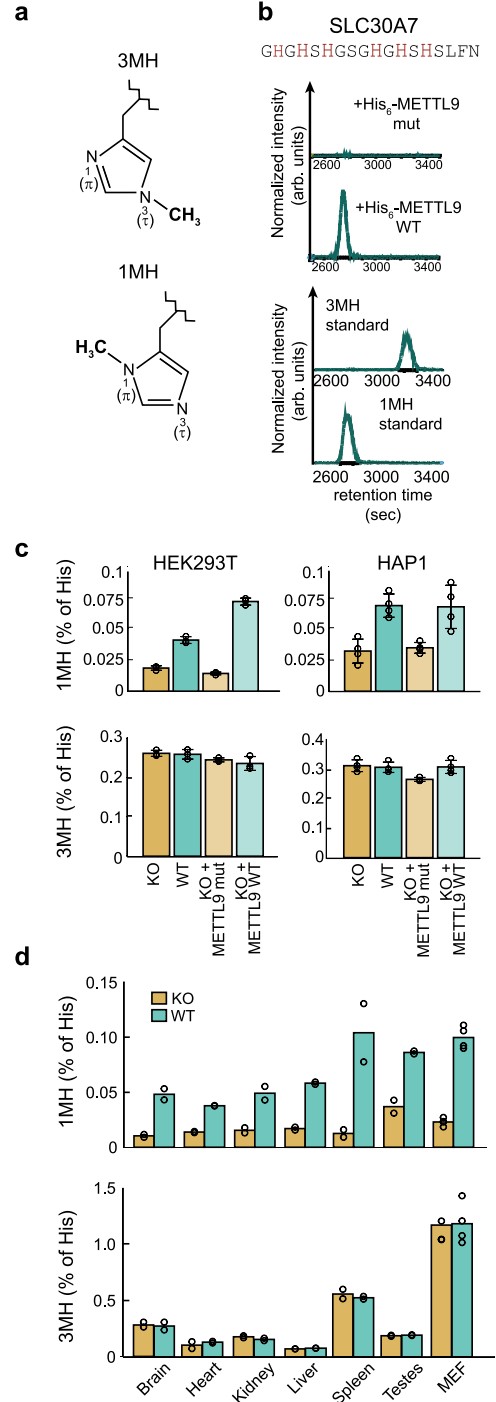

**Fig. 2 METTL9 generates pervasive 1-methylhistidine in vitro and in vivo. a** 3-Methylhistidine (3MH, top) and 1-methylhistidine (1MH, bottom). Polypeptide backbone indicated by jagged line. **b** Methylhistidine analysis of METTL9-treated SLC30A7 peptide. Shown are normalized LC/MS chromatograms of His$_6$-hMETTL9-treated samples subjected to acid hydrolysis, as well as of methylhistidine standards. **c** Proteinaceous 3MH and 1MH in HEK293T and HAP1 cells. WT (teal bars), *METTL9* KO (KO, orange bars), KO complemented with WT (KO + METTL9 WT, light teal) or mutant METTL9 (KO + METTL9 mut, light orange). Note that for HAP1 cells, the complementation is with hMETTL9-3xFLAG and the inactivating mutation is E174A, whereas for HEK293 cells, the complementation is with mMETTL9-HA and the inactivating mutation is D151K/G153R. Mean ± s.d ($n = 3$ (HEK293) or 4 (HAP1) biologically independent samples). **d** Proteinaceous 3MH and 1MH in WT or *Mettl9* KO mouse tissues and MEF cells. Mean with individual values of duplicates (tissues) or triplicates (MEF). Source data are provided as a Source Data file.

we generated the E. coli-expressed recombinant GST-tagged protein, as well as corresponding mutants where the relevant HxH motifs had been disrupted. Similar to what we already observed for His$_6$-ARMC6, all three proteins were methylated by GST-hMETTL9 in vitro, assessed using [$^3$H]AdoMet and fluorography (Fig. 4a). Importantly, methylation was abolished by the HxH-disrupting mutations in all cases (Fig. 4a). In conclusion, we have identified a total of 30 different human proteins that are targeted by METTL9 in vitro in the context of peptides and/or full-length recombinant proteins (summarized in Table 1).

**Identification of in vivo substrates of METTL9.** Prior to our work, HxH sequences from NDUFB3 and S100A9 were already reported to be histidine-methylated in vivo, and we have here shown that these sequences are methylated by METTL9 in vitro. Taken together with the high levels of METTL9-dependent 1MH in mouse and human proteomes, this strongly indicates that extensive METTL9-mediated methylation of HxH motifs occurs in vivo. To firmly establish this, we set out to directly demonstrate cellular METTL9-mediated methylation of specific in vitro substrates.

We generally found it challenging to assess the methylation status of the histidine-rich METTL9 substrates by protein MS, and relatively large amounts of pure material were required. However, we were able to isolate S100A9 from mouse peritoneal exudate neutrophils and, by matrix-assisted laser desorption-ionization (MALDI) MS, assess the methylation status of a peptide corresponding to the METTL9 target sequence (Fig. 4b, Supplementary Fig. 9a). Importantly, we found that this HGHSH-containing (Fig. 1d) peptide was completely unmethylated in *Mettl9* KO neutrophils, whereas it appeared exclusively in the mono- and dimethylated forms in WT neutrophils (Fig. 4b, Supplementary Fig. 9a). In the case of NDUFB3, over-expression of the FLAG-tagged protein in HEK293T cells enabled us to assess its in vivo methylation status. Importantly, the peptide corresponding to the N-terminal HxH-rich portion of the protein appeared as a mixture of mono-, di- and non-methylated forms in WT cells, whereas it was found exclusively in the unmethylated state in *METTL9* KO cells (Fig. 4c, Supplementary Fig. 9b). These results firmly establish that METTL9-mediated methylation of HxH-motifs occurs in vivo.

To obtain evidence for in vivo methylation of additional METTL9 substrates, we also attempted both targeted proteomics (using parallel reaction monitoring) on total cellular protein and

SLC30A1 and SLC39A6, and cyclin CCNT1, and they show a large variation in sequence and the number of HxH motifs present (from 1 to 4) (Fig. 3f). However, they all contained at least one ANGST HxH motif, whereas most of the peptides that were not methylated lacked such motifs (Fig. 3d and Supplementary Table 1).

Since we had mainly used peptide substrates when studying the in vitro activity of METTL9, we also set out to study methylation in the context of corresponding recombinant proteins. This was particularly relevant in the case of NDUFB3, for which HxH-methylation was reported in vivo, but where we failed to observe METTL9-mediated methylation of the corresponding peptide. For three proteins, i.e. NDUFB3, as well as CCNT1 and DNAJB12, both of which were efficiently methylated as peptides,

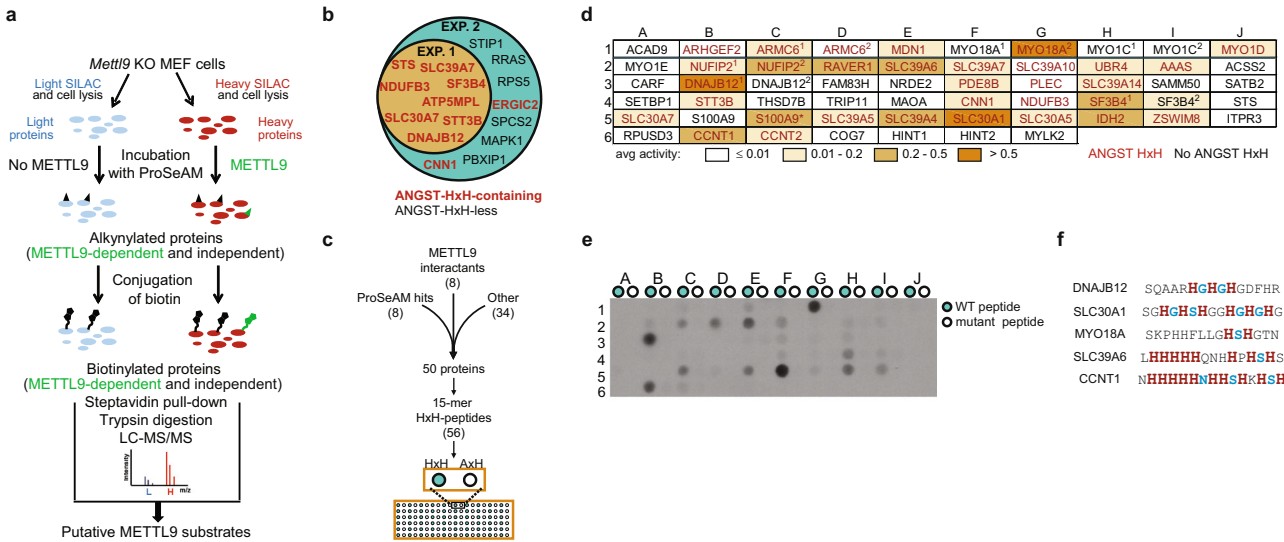

**Fig. 3 Identification of new cellular and peptide substrates of METTL9. a** Outline of the ProSeAM (propargylic Se-adenosyl-ʟ-selenomethionine) assay for identification of novel METTL9 substrates in cell extracts. **b** Venn diagram of mMETTL9 substrates identified in the two ProSeAM experiments. HxH-containing proteins are in red. **c** Schematic of array design for METTL9 substrate assessment. **d** Relative average MTase activity (from two independent arrays) presented in shades of orange, with gene names of relevant peptides indicated (ANGST HxH-containing in red). Superscripts denote multiple peptides derived from the same protein. All peptides are derived from human proteins, except mouse S100A9 (asterisk). Note that the array also contained an additional, HxH-less, peptide, based on the previously reported histidine-methylated sequence from MYLK2. **e** Representative result of METTL9-mediated methylation of a peptide array designed as in **c** and with each coordinate (e.g. A1) representing WT (HxH-containing; teal) and control (HxH-less; white) versions of a unique sequence (see also Supplementary Table 1). The experiment was performed twice with similar results. **f** Sequences of the top five METTL9 substrates from the peptide array, indicating ANGST residues (blue) within HxH motifs. Source data are provided as a Source Data file.

purification of select substrates by immunoprecipitation, but unfortunately these efforts were largely unsuccessful. However, one exception was DNAJB12, which was identified as a METTL9 target in both ProSeAM experiments, and found to be an excellent in vitro substrate (Fig. 3, Fig. 4a). We detected dimethylation of the relevant HSHSH-sequence in DNAJB12 immunoprecipitated from WT HEK293 cells (Supplementary Fig. 9c, d), but were, unfortunately, unable to detect the corresponding peptide in the KO cells. Still, this result suggests that DNAJB12 is an in vivo target of METTL9.

As we showed that *METTL9* KO substantially diminished global 1MH levels in mammalian proteomes (Fig. 2c, d), we also set out to directly demonstrate the presence of METTL9-dependent 1MH in a single cellular protein. For this purpose, we selected the zinc transporter SLC39A7, which carries a high number (32) of ANGST HxH motifs, and is relatively abundant. SLC39A7 was immunoprecipitated from HEK293T cells, and then subjected to acid hydrolysis and amino acid analysis. Strikingly, 4.4% of total histidine was found as 1MH in the immunoprecipitated SLC39A7 (Fig. 4d), i.e. a level ~100 fold higher than that found in the total HEK293T proteome, and corresponding, on average, to ~2.5 1MH modifications per SLC39A7, which contains 57 histidines. Importantly, 1MH in SLC39A7 was virtually absent in *METTL9* KO cells, but complementation of the KO cells with enzymatically active mMETTL9 restored 1MH to levels even higher than those observed in SLC39A7 from WT cells (Fig. 4d).

The above results directly demonstrate METTL9-dependent in vivo methylation for three proteins, namely S100A9, NDUFB3, and SLC39A7. In addition, we showed that METTL9 target sequences in ARMC6 and DNAJB12 are methylated in vivo, and high-throughput analysis of histidine methylation provided similar evidence for CCNT2 (shown to be a peptide substrate in Fig. 3d, e) and SLC39A7[16]. In summary, our data provide clear evidence that METTL9 is a broad-specificity MTase that

introduces 1MH at ANGST HxH motifs in numerous mammalian proteins, and the underlying in vitro (MTase assays) and cellular (MS detection of modification) findings have been summarized in Table 1.

**Effects of METTL9-mediated methylation on mitochondrial respiration and zinc binding.** NDUFB3 is an accessory subunit of the mitochondrial Complex I, and contains several methyl-histidines within a stretch of alternating histidines in its N-terminal region (Fig. 5a)[14]. We found that the presence of alternating histidines close to the N-terminus was a conserved feature of vertebrate NDUFB3, and that the number of HxH motifs showed an interesting variation between different organisms (Fig. 5a), indicating that methylation of these alternating histidines is functionally important. Thus, we investigated mitochondrial respiration in isolated mitochondria from HAP1 and in permeabilized HEK293T cells; in both cases ablation of METTL9 led to a reduction in oxygen consumption (Supplementary Fig. 10). The difference was particularly notable for so-called State 3 respiration, i.e. when respiration fuels ATP synthesis. A difference was only observed when respiration was driven by Complex I (of which NDUFB3 is a component), but not via Complex II (Supplementary Fig. 10). In addition, when WT, but not mutant, mMETTL9 was reintroduced into the KO HEK293T cells, respiration was restored to WT levels, showing that METTL9 enzymatic activity is required for optimal Complex I function (Fig. 5b). Taken together, these results indicate that METTL9-mediated methylation of NDUFB3 enhances Complex I-mediated mitochondrial respiration.

Many of the His residues targeted by METTL9 are involved in the binding of zinc and other metals (e.g. in S100A9 and zinc transporters)[13,25], suggesting that methylation may modulate metal binding to histidines. To investigate this, we used isothermal titration calorimetry (ITC) to measure zinc binding

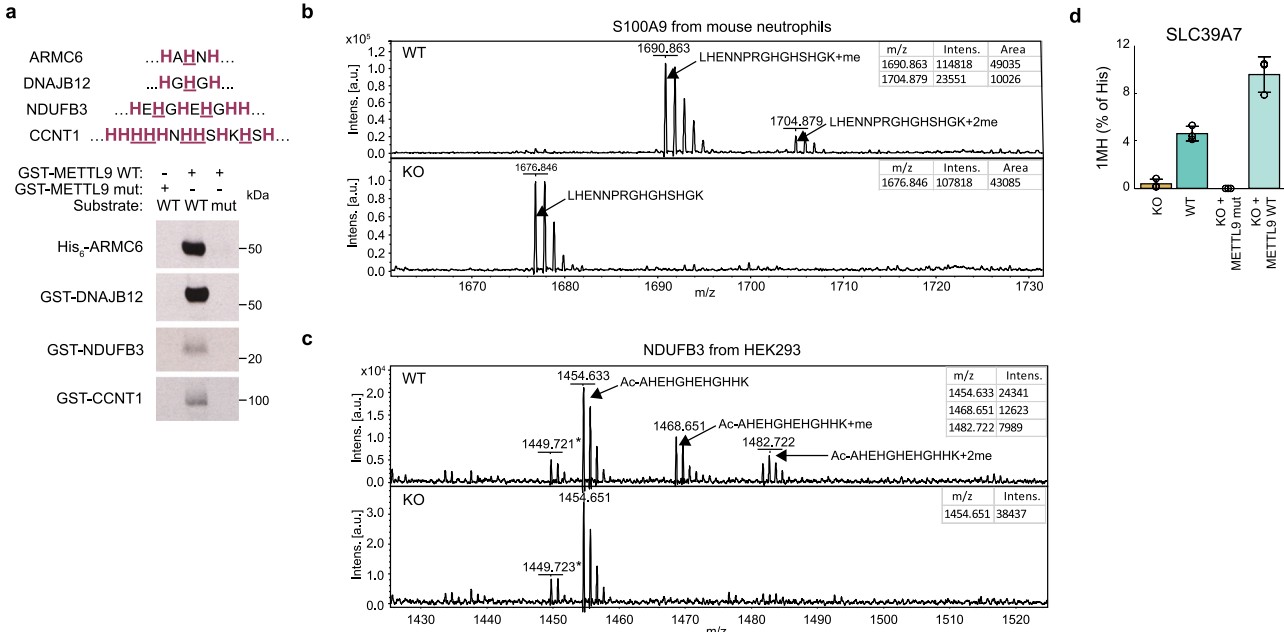

**Fig. 4 Confirmation of METTL9-dependent protein histidine methylation in vitro and in vivo. a** HxH-containing sequences of protein substrates with histidines marked in red and histidines mutated to arginines in the mutant substrates underlined (top). Fluorography showing in vitro activity of GST-hMETTL9 WT or E174A mutant (METTL9 mut) on the indicated proteins, either WT or with all HxH motifs removed by His-to-Arg mutations (mut) (bottom). 4 days exposure for ARMC6 and DNAJB12, 10 days exposure for CCNT1 and NDUFB3. **b** MALDI MS analysis of S100A9-derived peptides from WT and *Mettl9* KO mouse neutrophils. **c** MALDI MS analysis of peptides from NDUFB3-FLAG immunoprecipitated from WT and *METTL9* KO HEK293T cells. **d** Cellular SLC39A7 contains 1MH. The 1MH content of SLC39A7 immunoprecipitated from various HEK293-derived cells was determined by amino acid analysis. The following cells were used: WT, *METTL9* KO (KO), KO complemented with WT C-terminally HA-tagged mMETTL9 (KO + METTL9 WT) or KO complemented with the corresponding D151K/G153R mutant (KO + METTL9 mut). Mean ± s.d (*n* = 3 biologically independent samples). Source data are provided as a Source Data file. For the MALDI MS experiments (**b** and **c**), a table showing theoretical vs experimental mass of relevant peaks (and ppm deviation), as well as MASCOT results showing peptide coverage, can be found as Supplementary Data 1 and 2, respectively.

**Table 1 Identified METTL9 substrates.**

| Substrate | In vitro activity | | Observed in vivo |
|---|---|---|---|
| | On rec. protein | On peptide | |
| AAAS | | X | |
| ARMC6 | X | X | X[a] |
| CCNT1 | X | X | |
| CCNT2 | | X | X[16] |
| CNN1 | | X | |
| DNAJB12 | X | X | X[a] |
| IDH2 | | X | |
| MDN1 | | X | |
| MYO18A | | X | |
| MYO1D | | X | |
| NDUFB3 | X | | X[14,a] |
| NUFIP2 | | X | |
| PDE8B | | X | |
| RAVER1 | | X | |
| S100A9[b] | | X | X[13,a,c] |
| SF3B4 | | X | |
| SLC30A1 | | X | |
| SLC30A5 | | X | |
| SLC30A7 | | X | |
| SLC39A14 | | X | |
| SLC39A4 | | X | |
| SLC39A5 | | X | |
| SLC39A6 | | X | |
| SLC39A7 | | X | X[16,a,c] |
| STT3B | | X | |
| UBR4 | | X | |
| ZSWIM8 | | X | |

[a]In the present study.
[b]Mouse protein.
[c]METTL9-dependent, on endogenous protein.

to an SLC39A7-derived peptide containing six consecutive ANGST HxH motifs, as well as to a fully methylated (except the first His, in accordance with the presumed product specificity of METTL9; see "Discussion") version of this peptide. Interestingly, methylation significantly reduced zinc binding (Fig. 5c, Supplementary Fig. 11), suggesting that METTL9-mediated methylation modulates the binding of zinc and other metals to its target proteins.

## Discussion

Our results define METTL9 as a protein 1MH MTase. We show that METTL9 methylates ANGST HxH motifs in numerous proteins/peptides, and that METTL9-mediated 1MH formation is abundant in mammalian cells and tissues. Moreover, we find that NDUFB3, a component of mitochondrial Complex I, is methylated by METTL9, and that the enzymatic activity of METTL9 promotes Complex I-mediated respiration.

Strikingly, we observed a substantial reduction in 1MH content in total protein from all nine *METTL9* KO cells/tissues tested (six tissues and one MEF cell line from mice, as well as two human cell lines), and METTL9-dependent 1MH was found to account for up to ~0.1% of total histidine (Fig. 2c, d). Based on this, one may perform a simple calculation to further illustrate the potential extent of METTL9-mediated 1MH. The human proteome consists of ~10 million residues (~20,500 unique proteins with an average size of ~460 residues), and ~2.5% of these are histidines[26,27]. Assuming that at least half the proteome (10,000 proteins) is expressed in a typical human cell or tissue, this corresponds to ~125,000 histidine residues[28]. If one also assumes that the frequency of METTL9-mediated 1MH is independent of

**a**

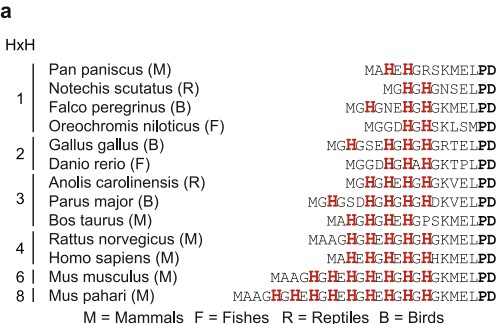

**b**

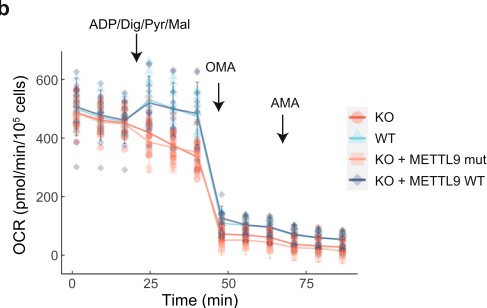

**c**

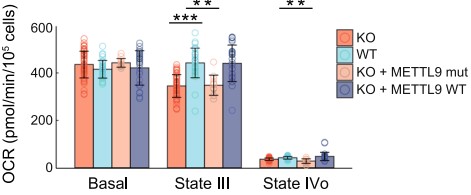

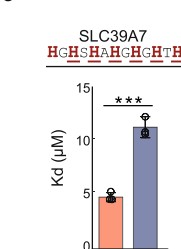

**Fig. 5 METTL9-mediated methylation affects mitochondrial respiration and reduces the binding affinity of a zinc transporter peptide to Zn$^{2+}$. a** Sequence alignment of the N-terminal end of NDUFB3 orthologues from various vertebrate species. Histidines are shown in red and the number of HxH indicated. Residues "PD" (in black, bold) are conserved in all vertebrates. **b** Oxygen consumption rate (OCR) of digitonin (Dig)-permeabilized WT (blue), *METTL9* KO (KO, red), *METTL9* KO complemented with WT (KO + METTL9 WT, purple) or D151K/G153R mutant mMETTL9-HA (KO + METTL9 mut, orange) HEK293 cells. Respiration mediated by Complex I was assessed under basal conditions, and after sequential addition of ADP/pyruvate (Pyr)/malate (Mal), oligomycin (OMA), and antimycin A (AMA). Shown are OCR traces (top) and quantification of respiratory states (bottom). Mean ± s.d. (n = 17 (KO), 11 (WT), 10 (KO + METTL9 WT), or 4 (KO + METTL9 mut) biologically independent samples). \*\*\*$P < 0.001$; \*\*$P < 0.005$ (state III: $P = 0.00372$, state IV: $P = 0.00157$); two-sided Dunnett's multiple comparison test. **c** Dissociation constant ($K_d$) between Zn$^{2+}$ and a synthetic SLC39A7 peptide that contains either none (Me0) or six 1MH (Me6) residues in an HxH context (methylated histidines underlined). Mean ± s.d. (n = 3 independent experiments). \*\*\*$P = 0.000451$; two-tailed Student's *t* test. Source data are provided as a Source Data file.

protein abundance, one can calculate that a 1MH frequency of 0.1% corresponds to ~125 stoichiometrically (100%) 1MH-modified histidine residues, and potentially a much higher number of sub-stoichiometrically modified residues. This agrees well with METTL9's relaxed sequence specificity (the human proteome contains 2807 instances of ANGST HxH[24]), and its ability to methylate the majority (~80%) of tested ANGST HxH-containing peptide substrates in vitro (Fig. 3d, e). So far, in vivo methylation of ANGST HxH motifs has been reported, by us and others, for six mammalian proteins (S100A9, NDUFB3, ARMC6, SLC39A7, DNAJB12, and CCNT2; Table 1) and we formally demonstrated METTL9-dependence for three of these (NDUFB3, S100A9, and SLC39A7). Future studies will likely focus on expanding the catalog of the METTL9's in vivo substrates.

Our work has firmly established that an HxH motif is an absolute requirement for METTL9-mediated methylation, as we found that disruption of the HxH motif invariably abrogated methylation in all tested substrates (28 peptides and 4 recombinant proteins). Moreover, we established, using three different peptide substrates, that a small residue (ANGST) is preferred as the middle residue, x, and this is also supported by other observations. First, ANGST HxH-motifs were present in all the proteins identified as METTL9 substrates in both of the two ProSeAM experiments (Fig. 3b), whereas this motif is found in only ~11% of mouse/human proteins[24]. Second, when HxH-containing candidate sequences were tested as METTL9 substrates (Fig. 3c–e) activity was observed for 27 out of the 32 ANGST HxH-containing peptides, but merely for one out of the 24 sequences that exclusively contained non-ANGST HxH motifs.

We demonstrated that an ANGST HxH motif is sufficient for METTL9-mediated methylation in most, but not all sequence contexts, as we failed to observe in vitro methylation of some (5 out of 32) of the tested ANGST HxH-containing peptide sequences. This was further corroborated by the observation that substitution of residues flanking the HxH motif in some cases abolished methylation (Supplementary Fig. 5). However, this flanking residue effect varied somewhat between the three sequences tested, and not all of the 5 ANGST HxH peptides that failed to be methylated contained a non-compatible flanking residue. Thus, additional sequence features also influence METTL9-mediated methylation of ANGST HxH motifs. In one case, NDUFB3, we observed METTL9-dependent methylation of the full-length protein both in vitro and in cells, but no in vitro methylation of the corresponding peptide. Conceivably, a structuring of the relevant NDUFB3 sequence, e.g. through interaction with other residues, may promote methylation in the context of the full-length protein.

We found that an HxH motif was invariably present in all METTL9 substrates, but many of the identified METTL9 substrates (e.g. ARMC6, S100A9, and NDUFB3) contained several consecutive HxH-motifs, typically HxHxH, and mutating an HxHxH-motif into HxH severely diminished methylation in several cases (Fig. 1e). This suggests that a stretch of more than one HxH may be a biologically more relevant substrate for METTL9. Interestingly, we have not observed in our MS/MS data the methylation of the first His residue in a stretch of alternating histidines, in agreement with previously reported methylation patterns of NDUFB3 and S100A9 (Fig. 1d)[13,14]. This may suggest that METTL9 can only methylate the second

histidine of the HxH motif, i.e. that all but the first residue in a stretch of alternating histidines have the potential to become methylated. In this context, zinc transporters of the SLC30A and SLC39A families are a very interesting group of METTL9 substrates. They are responsible for transporting zinc and other metals across cellular membranes, but many of the 23 human proteins remain poorly characterized[25]. They contain a high number of metal-binding, ANGST-enriched alternating histidines: eleven of these proteins collectively account for 24% of the instances of ANGST HxHxH-motifs in the human proteome (but for only 6.5% of total HxHxH[24]). Interestingly, the METTL9-targeted residues in S100A9 are also involved in metal coordination[13]. Indeed, we showed that 1MH modification of a SLC39A7-derived stretch of alternating histidines caused a reduction in zinc binding (Fig. 5c), and it may be of interest to further investigate the potential role of METTL9 in regulating metal ion homeostasis.

The *Mettl9* KO mice were without apparent phenotype, and this somewhat contrasts with the dramatic reduction in 1MH observed proteome-wide. However, it is not unprecedented that a KO mouse devoid of a modification enzyme shows a strong molecular phenotype but no other, easily observable, phenotype. For example, mouse KO of the tRNA modification enzyme Alkbh8 caused mismodification of several distinct tRNA iso-acceptors at the critical wobble uridine position, but gave no overt phenotype[29,30]. However, later studies showed defects in seleno-protein synthesis and stress tolerance in the *Alkbh8* KO mice[31,32], and it was recently reported that homozygous inactivating mutations in the human *ALKBH8* gene causes intellectual dis-ability and general developmental delay[33]. Likewise, more detailed studies, e.g. on metal biology and mitochondrial meta-bolism, will be required to elucidate the biological function of METTL9.

Our in vitro methylation experiments (Fig. 1a, Supplementary Fig. 2a, c) demonstrated considerable methylation by recombi-nant METTL9 in WT extracts, indicating substantial hypo-methylation of METTL9 substrates. This observation indicates that, rather than being static, METTL9-mediated methylation is dynamic and of potential regulatory importance. It would therefore be intriguing to explore the dynamics of methylation, and whether it can be reversed or modulated by specific deme-thylases and reader proteins, in a manner analogous to lysine methylation on histones[34,35]. Moreover, we also observed sub-stantial METTL9-independent 1MH in all examined cells and tissues, and the corresponding MTases remain to be identified, e. g. the one introducing 1MH (in a non-HxH context) in MYLK2.

Note: Since no other enzymes have previously been reported to generate 1MH in proteins, METTL9 cannot readily be given any EC (Enzyme Commission) number, see e.g. the ENZYME data-base[36]. "EC 2.1.1.85 Protein-histidine N-methyltransferase" already exists, but only encompasses enzymes that generate 3MH. Thus, we recommend that a new entry is made for "EC 2.1.1.xxx Protein-histidine N1-methyltransferase", and that EC 2.1.1.85, also for the future, is restricted to protein-histidine N3-methyltransferases.

## Methods

**Cloning and mutagenesis**. ORFs were amplified with Phusion High-Fidelity DNA polymerase (Thermo Fisher Scientific) from HeLa or HEK293 cDNA. Primers were designed to generate PCR products flanked by sequences com-patible with the appropriate vectors for In-Fusion Cloning (Takara Bio). Mutagenesis was performed using mutagenic primers designed with the PrimerX program and the QuickChangeII Site-directed mutagenesis kit (Agilent) or by overlap extension PCR. All constructs were sequence-verified. Human NDUFB3 cDNA was amplified from a HEK293T cDNA library, and the full length DNA fragment was inserted into either pQCXIH or pcDNA3 vector with C-terminal FLAG tag.

Mouse Mettl9 cDNA was amplified from Fantom clone (AK075586), and the full length DNA fragment was inserted into pcDNA3 vector with C-terminal FLAG or HA tag (pcDNA3-mouse (m)METTL9-FLAG, pcDNA3-mMETTL9-HA). For baculovirus expression system, N-terminal truncated mMETTL9 (residues 22–318) was cloned into the pFastBac HT vector, which encodes a tobacco etch virus (TEV) protease recognition sequence inserted after an N-terminal polyhistidine tag. The oligonucleotides used for cloning, mutagenesis, and for CRISPR-Cas9-mediated gene knock-out are listed in Supplementary Table 2.

**Expression and purification of recombinant proteins**. GST-tagged proteins GST-hMETTL9, GST-hMETTL9 E174A, GST-SLC39A7 (31-137aa), GST-DNAJB12 (1-243 aa), GST-DNAJB12 H185R, GST-NDUFB3, GST-NDUFB3 H5R/H9R, GST-CCNT1 and GST-CCNT1 H519R/H520R/H523R/H524R/H528R were expressed in *Escherichia coli* and purified using Glutathione Sepharose 4B (Sigma-Aldrich). His$_6$-tagged proteins His$_6$-hMETTL9, His$_6$-hMETTL9 E174A, His$_6$-ARMC6, His$_6$-ARMC6 H261R, His$_6$-ARMC6 A262G, His$_6$-ARMC6 H263R, His$_6$-ARMC6 N264D, His$_6$-ARMC6 H265A, His$_6$-ARMC6 A266G, and His$_6$-ARMC6 K267R were expressed in *E. coli* and purified using Ni-NTA-agarose (Qiagen). In sum-mary, the appropriate pGEX-6P-2 or pET28a constructs were transformed into the chemically competent BL21- CodonPlus (DE3)-RIPL (Agilent Technologies) *E. coli* strain. The bacteria were cultured overnight, inoculated into a larger volume of TB medium with appropriate antibiotics, and after the OD600 reached ~0.8, protein expression was induced with 0.1 mM isopropyl β-D-1-thiogalactopyranoside (IPTG), at 18 °C, 250 rpm, followed by overnight growth. For His-tagged proteins, bacteria were lysed in His-Lysis Buffer (50 mM Tris-HCl pH 8.0, 500 mM NaCl, 5% (v/v) glycerol and 1% (v/v) Triton X-100) supplemented with 1 mg/ml lyso-zyme, 5 mM 2-mercaptoethonol, 10 mM imidazole, and 1x cOmplete (EDTA-free) Protease Inhibitor Cocktail (Roche) and purified using Ni-NTA-agarose (Qiagen), according to the manufacturer's instructions. For GST-tagged proteins, bacteria were lysed in GST-Lysis Buffer (50 mM Tris-HCl pH 7.6, 500 mM NaCl and 0.5% Triton X-100) supplemented with 1 mg/ml lysozyme, 1 mM DTT, and 1x cOm-plete Protease Inhibitor Cocktail (Roche) and purified using Glutathione-Sepharose 4 B (Sigma-Aldrich). Eluted proteins were buffer exchanged to Storage Buffer (50 mM Tris-HCl pH 8.0, 300 mM NaCl, 5% glycerol and 1 mM DTT) using cen-trifugal concentrators with a molecular weight cut-off of 10 kDa (Sartorius) and stored at −80 °C.

For the ProSeAM assay, mMETTL9 was expressed in baculovirus-infected Sf9 insect cells. Baculovirus was generated using the Bac-to-Bac baculovirus expression system (Invitrogen) according to the manufacturer's instruction. The baculovirus-infected cells were harvested 48 h after transfection, and cell pellets were stored at −80 °C until purification. Frozen Sf9 cells were resuspended in 20 mM Tris-HCl buffer (pH 8.0) containing 500 mM NaCl, 0.1% NP-40, 10% glycerol, 20 mM imidazole, DNase I (Sigma Aldrich) and cOmplete (EDTA-free) Protease Inhibitor Cocktail (Roche). Cells were lysed by sonication and clarified by centrifugation. The cell lysate was loaded onto a HisTrap HP column (GE Healthcare), and eluted with 50 mM Tris-HCl buffer (pH 8.0) containing 500 mM NaCl, 10% glycerol and 500 mM imidazole. After exchanging the buffer using a HiTrap Desalting column (GE Healthcare), the N-terminal polyhistidine tag was cleaved by incubating with TEV protease overnight at 4 °C. Then, the cleaved mMETTL9 protein was reapplied to a HisTrap HP column, and the flow-through fraction was collected. The mMETTL9 protein was further purified by size-exclusion column chromatography using a HiLoad Superdex 200 16/60 (GE Healthcare), and then the buffer was exchanged to 20 mM Tris-HCl buffer (pH 8.0) containing 150 mM NaCl and 20% glycerol using a HiTrap Desalting column. Purified protein was concentrated using an Amicon Ultra-15 centrifugal filter unit (Millipore, 10 kDa MWCO) to 0.4 mg/ml and flash frozen in liquid nitrogen.

**In vitro methyltransferase assays on recombinant proteins and protein extracts**. In vitro methyltransferase assays were performed using Reaction Buffer (50 mM Tris-HCl pH 7.5, 50 mM NaCl, 5 mM EDTA) with 0.64 μM $^3$H-AdoMet (Perkin–Elmer, specific activity = ~77–78 Ci/mmol) or 1 mM unlabeled AdoMet for MS analysis of samples. For methylation of recombinant His$_6$-ARMC6 WT and mutant proteins or the GST-SLC39A7 (aa 31–137) fragment, reactions con-tained 10 μg substrates and 1 μM WT or E174A mutant His$_6$-hMETTL9. For methylation of subcellular protein extracts, 100 μg protein from HAP1 fractions isolated using the Subcellular Protein Fractionation Kit for Cultured Cells (Thermo Scientific) were used as substrates, with 1 μM WT or E174A mutant GST-hMETTL9 enzymes. For the recombinant substrate panel, 2 μM His$_6$-ARMC6 WT and H263R mutant, GST-DNAJB12(1-243 aa) WT and H185R mutant, or GST-NDUFB3 WT and H5R/H9R mutant, or 1 μM GST-CCNT1 WT and H519R/H520R/H523R/H524R/H528R mutant were used as substrates with 1 μM GST-hMETTL9 WT or E174A mutant enzyme. Reactions were incubated at 36 °C for 1 h, and stopped by the addition of NuPAGE loading buffer. Proteins were separated by SDS-PAGE. For fluorographic analysis, the proteins were then transferred to PVDF membranes, stained with Ponceau S, dried, sprayed with a scintillation solution (57.5% 2-methylnaphthalene, 40% pentylacetate, 2.5% diphenyloxazole) and exposed to Eastman Kodak Co. Bio-Max MS film (Sigma-Aldrich) at −80 °C.

**Methylhistidine content analysis of cultured cells, mouse tissues and peptides**. HAP1 cells, MEFs, and mouse tissues were lysed in RIPA buffer (Sigma-Aldrich), followed by precipitation with 10% (v/v) aqueous trichloroacetic acid for 20 min on ice and centrifugation (16,000 × g, 15 min, 4 °C). The pellets were washed with cold isopropanol, pelleted again, and heated to 95 °C to remove residual isopropanol. The dry pellets were transferred to vacuum hydrolysis tubes (Thermo Fisher Scientific), and 200 µl Sequencing Grade 6 M HCl (Thermo Fisher Scientific) was added. The pellets were hydrolyzed under vacuum at 110 °C for 48 h. The samples were then desiccated at 60 °C and the pellet resuspended in 800 µl of $H_2O$. These solutions were filtered through 0.22 µm syringe filters (Millex-GP) to remove insoluble material and the liquid was again evaporated at 70 °C. The LC-MS/MS analysis to quantify methyl-histidine content was performed by BEVITAL AS.

Peptides derived from SLC30A7 (aa 163-180), CCNT1 (aa 515-531), and ARMC6 (aa 253-267) purchased from Peptide2.0 were methylated with 1 µM His6-hMETTL9 WT or His6-METTL9 E174A, and 1 mM AdoMet in Reaction Buffer (50 mM Tris pH 7.5, 50 mM NaCl, 5 mM EDTA). All peptides had C-terminal amides. The samples were then subjected to acid hydrolysis. Briefly, liquid samples were dried in a Speed Vacuum (Thermo Fisher) and put under $N^2$ in glass vials. 1 mL 6 M HCl was added to the bottom of the reaction vials and the samples were heated to 110 °C for 36 h. The samples were then placed in clean reaction vials and heated for a further 5 h, then desiccated under vacuum for 16 h to remove remaining acid. The samples were analyzed using a ACCQ TAG ULTRA C18 1.7 µM column and the Xevo-G2S LC-MS machine (Waters) and analyzed with MassLynx V4.1 software. 1-Methyl-L-histidine (Sigma-Aldrich) and 3-Methyl-L-histidine (Sigma-Aldrich) were used as 3MH and 1MH standards, respectively. Note the different nomenclature for methylhistidine used in biochemistry and chemistry due to different systems of numbering the atoms in the imidazole ring.

For methylhistidine content analysis of HEK293 cells, the cell pellets precipitated with acetone were hydrolyzed with 6 N HCl at 110 °C for 24 h, and the extracted amino acids were dissolved in 25 µL of 5 mM $HCOONH_4$/0.001% formic acid. The Extracted samples were applied to a liquid chromatograph (Vanquish UHPLC; Thermo Fisher Scientific) coupled to a triple quadrupole mass spectrometer (TSQ Vantage EMR; Thermo Fisher Scientific). The amino acids derived from the protein samples were separated on a C18 column (YMC-Triart C18, 2.0 × 100 mm length, 1.9 µm particle size; YMC.CO., LTD., Kyoto, Japan). Mobile phase A was comprised of 5 mM ammonium formate with 0.001% formic acid, and mobile phase B was comprised of acetonitrile. Following different slopes were used for a gradient elution at a flow rate of 0.3 mL/min: 0/0 – 1.5/0 – 2/95 – 4/95 – 4.1/0 – 7/0 (min/%B). The effluent from the column was directed to an electrospray ion source (HESI-II; Thermo Fisher Scientific) connected to the triple quadrupole mass spectrometer operating in the positive ion multiple reaction monitoring mode, and the intensity of specific MH + → fragment ion transitions were recorded (His m/z 156.1 → 83.3; 93.2; 110.2, 1MH 170.1 → 95.3; 97.3; 109.2 and 3MH m/z 170.1 → 81.3; 83.3; 124.2). The electrospray conditions were; spray voltage of 3000 V, vaporizer temperature of 450 °C, sheath gas pressure 50 arbitrary units, auxiliary gas pressure 15 arbitrary units, and collision gas pressure 1.0 mTorr. With each batch of experimental samples a series of standard samples was simultaneously prepared and run. Calibration curves were constructed for 1MH, 3MH, and His from the data obtained from the standard samples with a range of 1 – 250 nM. The measured concentration and percentage of 1MH and 3MH in each experimental sample was calculated from the calibration curves.

**SLC39A7 methylhistidine content analysis**. Pelleted HEK293T cells (2 mg) were dissolved in 500 µL of lysis buffer (50 mM pH7.5 Tris-HCl, 400 mM NaCl, 0.1% NP40), and sonicated for 10 s on ice. After centrifugation (18,000 × g for 10 min), the supernatant was collected in a new tube. To immunoprecipitate SLC39A7, 1 µg of anti-ZIP7 antibody (Protein Tech, #19429-1-AP) was added to the cell lysate, and incubated for 1 h at 4 °C with a gentle agitation. For antibody alone control, 1 µg of anti-ZIP7 antibody was also added to 500 µL of the lysis buffer without cell lysates. Ten µL of Protein A/G PLUS-Agarose (Santa Cruz, SC-2003) was added in the tube, and incubated for 1 h at 4 °C with a gentle agitation. The agarose beads were washed three times with PBS. The bound proteins were separated with SDS-PAGE and transferred to a PDVF membrane. The PVDF membrane was stained with Coomassie Brilliant Blue R250, and the band corresponding to SLC39A7 (around 50 kDa) was excised. The membrane bound proteins were hydrolyzed and quantitated as described above for HEK293 cells. Each amino acid content value was subtracted with that of the antibody alone control, since molecular weight of SLC39A7 and the heavy chain of anti-ZIP7 antibody was almost the same and the excised band around 50 kDa consisted of two species.

**Generation of human METTL9 KO cell lines**. HAP-1 METTL9 KO cells were generated as a custom project by Horizon Genomics. The METTL9 gene was disrupted by the CRISPR-Cas9 method using a guide RNA targeted towards exon 2, resulting in an 8 bp frameshift mutation. The METTL9 KO cell line is currently commercially available at Horizon, catalog ID HZGHC004343c010. The HAP1 KO cells were complemented with C-terminally 3xFLAG-tagged WT or E174A mutant hMETTL9 by stable transfection with the appropriate p3xFLAG-CMV-14 constructs using the FuGENE6 Transfection Reagent (Roche), followed by selection in medium containing 1 mg/ml geneticin (Gibco). Individual clones were screened by

western blot for the presence of the tag using anti-FLAG antibodies (Sigma-Aldrich, F1804). WT T-Rex Flp-In 293 cell lines inducibly expressing C-terminally GFP-tagged WT METTL9 were generated according to protocol (Thermo Fisher Scientific). METTL9 KO HEK293T cells were generated by deletion of Exon 3 with the CRISPR-Cas9 system. hCas9 and two guide RNA expression vector, PX330-B/B were kindly provided by Dr. T. Hirose[37]. Two guide RNAs targeted to introns of upstream (5′-TAATAAGTGATTATGGTTGT-3′) and downstream (5′-GCAG-TATTTTTCTGGAGCGG-3′) of exon 3 in METTL9 were cloned into BbsI and BsaI site of PX330-B/B vector, respectively. The plasmid (PX330-B/B-gMETTL9) was then transfected into HEK293T cells together with pEGFP-C1. Two days after the transfection, GFP positive cells were sorted and single cells were seeded onto 96 well plate. The KO clones were screened by genomic PCR.

For the generation of HEK293T KO cells complemented with C-terminally HA-tagged WT or D151K/G153R mutant mMETTL9, plasmids for retroviral expression (pQCXIP-mMETTL9-HA or pQCXIP-mMETTL9-DKGR-HA) were transfected into retrovirus packaging cells using PEI transfection reagent (Polysciences, Inc.). 24 h after the transfection, the virus containing culture supernatant were transferred to METTL9 KO HEK293T cells with 4 µg/mL of polybrene. 24 h after the infection, 1 µg/mL puromycin were added, and the cells cultured for 2 weeks.

**Generation of Mettl9 KO mice**. The Mettl9 KO mice were generated by the CRISPR-Cas9 system. Guide RNAs that target exon 3 of mouse Mettl9 were screened with CRISPR design tools (http://crispr.mit.edu/). The T7 promoter sequence 5′-TTAATACGACTCACTATAGG-3′ and the 20-mer Mettl9 guide RNA sequence 5′-GAGACTGCTTAGAATTAATC-3′ were cloned into the AflII site of the gRNA Cloning Vector, a gift from George Church (Addgene #41824)19. Oligos Insert-F (5′-TTTCTTGGCTTTATATATCTTAATACGACTCACTATAGG AGACTGCTTAGAATTAATC-3′) and Insert-R (5′-GACTAGCCTTATTT-TAACTTGCTATTTCTAGCTCTAAAACGATTAATTCTAAGCAGTCTC-3′) were annealed and extended to construct a 100-bp double-stranded DNA fragment using Phusion polymerase (New England Biolabs, Japan). The gRNA Cloning Vector was linearized with AflII and the inserts were incorporated via Gibson assembly (NEB, Japan, E2611S). To prepare hCas9 mRNA and Mettl9 guide RNAs, an in vitro transcription reaction kit with T7 RNA polymerase (mMESSAGE mMACHINE T7 Ultra Kit, Life Technologies) was used; the transcribed RNAs were purified with the MEGAclear Kit (Life Technologies). The quality of the guide RNA was checked by an in vitro Cas9 cleavage assay with recombinant hCas9 nuclease (NEB) and a 400 bp PCR product around the guide RNA target site. The Cas9 mRNA and guide RNA were injected into fertilized eggs (mouse strain C57BL/6 J), and the mutated Mettl9 gene versions were screened by PCR after BsrI digestion, because the BsrI site is located after the guide RNA target site. One in 8 heterozygous (+/−) males carried the expected 16 bp deletion. The heterozygous mouse was crossed with a WT C57BL/6 J mouse, and the male (+/−) and female (+/−) mice were crossed to obtain Mettl9 KO (−/−) mice and WT (+/+) mice. Animals were maintained on a 12-h light/dark cycle with access to food and water ad libitum. The temperature and humidity were maintained at 22–23 °C and 50–60%, respectively. Animal health was checked by the animal facility staff 5 times per week. All experiments involving mice complied with all relevant ethical regulations for animal testing and research, and were carried out according to protocols approved by the Animal Experiment Committee of the RIKEN Center for Brain Science.

**MS detection of methylation on recombinant proteins**. Samples were run on NuPAGE 4–12% Bis-Tris protein gels in MES buffer (ThermoFisher), stained with SimplyBlue (Invitrogen) and destained in water. Proteins were excised from the gel and subject to in-gel digestion by trypsin or chymotrypsin. The dried peptides were dissolved in 10 µL of aqueous 2% acetonitrile containing 1% formic acid and 5 µL of the sample was injected into a Dionex Ultimate 3000 nano-UHPLC system (Sunnyvale, CA, USA) coupled online to a Q Exactive mass spectrometer (ThermoScientific, Bremen, Germany) equipped with a nano-electrospray ion source. For liquid chromatography separation, an Acclaim PepMap 100 column (C18, 3 µm beads, 100 Å, 75 µm inner diameter, 50 cm) was used. A flow rate of 300 nL/min was employed with a solvent gradient of 2–30% B in 60 min. Solvent A was 0.1% formic acid and solvent B was 0.1% formic acid in 90% acetonitrile. The mass spectrometer was operated in the data-dependent mode to automatically switch between MS and MS/MS acquisition. Survey full scan MS spectra (from m/z 400 to 2000) were acquired with the resolution R = 70,000 at m/z 200, after accumulation to a target of 1e6. The maximum allowed ion accumulation times were 100 ms. The sequential isolation of up to the ten most intense ions, depending on signal intensity (intensity threshold 5.6e3) were considered for fragmentation using higher-energy collision-induced dissociation (HCD) at a target value of 100,000 charges and a resolution R = 17,500 with normalized collision energy (NCE) 28. Target ions already selected for MS/MS were dynamically excluded for 30 s. The isolation window was m/z = 2 without offset. Data were analyzed with in-house maintained human protein database using SEQUEST™ and Proteome Discoverer™ (Thermo Fisher Scientific). The mass tolerances of a fragment ion and a parent ion were set as 0.05 Da and 10 ppm, respectively. Methionine oxidation and cysteine carbamido-methylation were selected as variable modifications.

**MS detection of ARMC6 methylation in cells.** hMETTL9-GFP was over-expressed in HEK-293-derived T-REx-Flp-IN cells and affinity purified using GFP-Trap beads. In brief, cells in a 6-well plate were induced with 1 mg/ml doxycyclin overnight to express the GFP fusion proteins. The cells were harvested and lysed in RIPA buffer containing cOmplete protease inhibitor cocktail, phenylmethylsulfonyl fluoride and sodium orthovanadate (Sigma-Aldrich) on ice for 20 min. After centrifugation (16,000 × g, 5 min, 4 °C), the cleared lysate was diluted in Dilution Buffer (50 mM Tris-HCl pH 7.5, 150 mM NaCl, 0.5 mM EDTA). Washed GFP-Trap beads (Chromotek) were incubated with the lysate with overhead rotation for two hours, collected by centrifugation (3000 × g, 3 min, 4 °C), washed two times with 500 µl of the Dilution Buffer and frozen at −80 °C. Proteins were eluted through on-bead proteolysis using trypsin[38], the resulting peptides desalted using StageTips[39] and analyzed by nLC-MS/MS using EASY-nLC 1200 coupled to a Q Exactive HF-X mass spectrometer (Thermo Scientific) operated in the data-dependent acquisition mode[40]. All MS files were analyzed using MaxQuant (v1.6.3.3)[41] using the default settings except for the following variable modifications: monomethylation (Lys, Arg and His), di-methylation (Lys and Arg), and tri-methylation (Lys). The data were searched against a database comprising the canonical isoforms of human proteins as downloaded from Uniprot (Uniprot Complete proteome: UP_2017_04\Human\UP000005640_9606.fasta).

**MS detection of NDUFB3 and S100A9 methylation in cells.** For NDUFB3, the plasmids for retrovirus expression of C-terminally FLAG-tagged NDUFB3 (pQCXIH-NDUFB3-FLAG) were transfected into retrovirus packaging cells using PEI transfection reagent (Polysciences, Inc.). 24 h after the transfection, the virus-containing culture supernatants were transferred to WT or *METTL9* KO HEK293T cells with 4 µg/mL of polybrene. 24 h after the infection, 500 µg/mL hygromycin B was added and cells were cultured for 2 weeks. The cells were harvested and lysed in IP buffer (1xPBS, 1% n-Dodecyl-β-D-maltoside, protease inhibitors). M2-agarose beads (Sigma-Aldrich) were added and incubated with the lysate for 1 h at 4 °C, then washed three times with PBS. The bound proteins were separated with SDS-PAGE, and the gel band corresponding to NDUFB3-FLAG was excised and in-gel digested with trypsin.

For S100A9 isolation, peritoneal exudate neutrophils (PEN) were prepared as described[42]. WT and *Mettl9* KO mice were intraperitoneally injected with 3% (w/v) proteose peptone (Difco Laboratories Inc.) (4 ml per mouse) twice (2^nd injection was after 12–15 h 1st injection) and sacrificed 2-3 h after the 2^nd injection of proteose peptone to collect PEN. PEN-enriched fractions were isolated from the harvested intraperitoneal cells with a Percoll gradient. PEN fraction (2 × 10^5 cells) was mixed with 2x Laemmli SDS-sample buffer, and separated with SDS-PAGE. The bands corresponding to S100a9 (observed as a major band at 14 kDa) were excised, and in-gel digested with *Achromobacter* protease I (API). The digested peptide fragments were analyzed with MALDI-MS.

**MALDI MS analysis of hNDUFB3 and mS100A9.** The SDS-PAGE-separated bands were excised and destained. The gel slices were reduced with 50 mM dithiothreitol and 4 M guanidine-HCl at 37 °C for 2 h, followed by alkylation with 100 mM acrylamide at 25 °C for 30 min. The NDUFB3 and S100A9 were digested with trypsin (TPCK-treated, Worthington Biochemical) and Achromobacter protease I (a gift from Dr. T. Masaki, Ibaraki University, Ibaraki), respectively. The reaction was carried out at 37 °C for 12 h in a digestion buffer (10 mM Tris-HCl (pH8.0)/0.03% n-Dodecyl-β-D-maltoside) An α-cyano-4-hydroxycinnamic acid (Brucker Daltonics)–saturated solution in 33% acetonitrile/67% water containing 0.1% trifluoroacetic acid was prepared. A 0.5 µl volume of each sample was mixed with 1.0 µl of the matrix solution and air-dried at room temperature on an MTP AnchorChip Targets plate (Brucker Daltonics). The MS and MS/MS spectra were acquired with a rapifleX MALDI Tissuetyper (Brucker Daltonics). The mass spectrometer was operated in the positive-ion mode and reflector mode the following high voltage conditions (Ion Source1:20.000 kV, PIE: 2.680 kV, Lens: 11.850 kV, Reflector 1: 20.830 kV, Reflector 2: 1.085 kV, Reflector 3: 8.700 kV). The MSMS spectra were obtained using LIFT method the following high voltage conditions (Ion Source1:20.000 kV, PIE: 2.680 kV, Lens: 11.850 kV, Reflector 1: 20.830 kV, Reflector 2: 1.085 kV, Reflector 3: 8.700 kV, Drift tube 1: 14.000 kV, Drift tube 2: 18.700 kV, MS/MS pulse: 2.600 kV).

The acquired data were processed using FlexAnalysis (version 4.0, Brucker Daltonics) and BioTools (3.2 SR5, Brucker Daltonics). The processed data were used to search with MASCOT (version 2.7.0, Matrix Science) against the in-house database including the amino acid sequences of NDUFB3 and S100A9, using the following parameters: enzyme = trypsin (NDUFB3) or Lys-C/P (S100A9); maximum missed cleavages = 1; variable modifications = Acetyl (Protein N-term), Oxidation (M), Methyl (H), Propionamide (C); product mass tolerance = ±50 ppm; product mass tolerance = ±0.5 Da (LIFT mode); instrument type = MALDI-TOF-TOF.

**MS detection of DNAJB12 methylation in cells.** HEK293 cells grown to 80% confluency in 150 mm plates were harvested and lysed in RIPA buffer containing cOmplete protease inhibitor cocktail and phenylmethylsulfonyl fluoride (Sigma-Aldrich) on ice for 20 min after brief sonication. The lysate was then cleared by centrifugation (16,000 × g, 5 min, 4 °C), diluted two-fold with Dilution Buffer (50 mM Tris-HCl pH 7.5, 150 mM NaCl, 0.5 mM EDTA) and incubated with

(0.5 µg/mg lysate) DNAJB12 antibody (16780-1-AP, Proteintech) with overhead rotation for two hours. Washed Protein A Agarose (Merck) was then added and incubated for 2 more hours. The resin was collected by centrifugation (3000 × g, 3 min, 4 °C), washed twice with Dilution Buffer, then twice with water, transferring the resin to a new low-protein binding tube with each wash, and finally frozen at −80 °C. Immunoprecipitated DNAJB12 was separated by SDS-PAGE and in-gel digested with Chymotrypsin (Roche)[43]. The resulting peptides were, without a desalting step, analyzed using a EASY-nLC 1200 ultrahigh-pressure system (Thermo Fisher Scientific) coupled to a Q Exactive HFX Orbitrap mass spectrometer (Thermo Fisher Scientific) operated in a data-dependent acquisition mode[40]. All raw MS files were analyzed with Max-Quant (version 1.6.0.17)[41] and searched against a database composed of the canonical isoforms of human proteins as downloaded from Uniprot in April 2017 (Uniprot Complete proteome: UP_2017_04/Human/UP000005640_9606.fasta) using the default setting with few exceptions. Methylation of lysine (mono, di, and tri), arginine (mono and di), and histidine (mono) were set as variable modifications and missed cleavages were restricted to one.

Ion chromatograms for the different methylated forms of a peptide covering amino acid 173–190 of DNAJB12, DQFGDDKSQAARHGHGHGDF, were extracted using Xcalibur Qual Browser, ver. 4.1 (Thermo Fisher Scientific). Selective ion settings for Asp171-Phe190 (z = 4) in Supplementary Fig. 9c were 546.24 (Me0), 549.75 (1xMe) and 553.25 (2xMe), 7 p.p.m.

**Generation and methylation of peptide arrays.** Peptide arrays of 15-mer peptides were generated using the SPOT method[44,45]. Methylation reactions were conducted by incubating the arrays in Reaction Buffer (50 mM Tris-HCl pH 7.8, 50 mM NaCl, 5 mM EDTA) with 0.76 µM [3H]-AdoMet (PerkinElmer) and 0.5-1 µM of His6-hMETTL9. For His-to-Ala replacements, single histidines from peptides from ARMC6 (aa 253–267), SLC39A5 (aa 367–381), and mouse S100A9 (aa 99-113, note the additional Cys-to-Ser substitution to reduce non-enzyme-catalyzed methylation) were substituted for alanines. For the middle and flanking HxH residue replacements, the peptides as above were additionally modified to contain only one HxH. The relevant residues were mutated to all proteinogenic amino acids except Trp and Cys. Note that for the middle residue replacements, the individual arrays were cut from single images and pasted together to follow the same order of mutations introduced.

For the candidate substrate array, HxH-containing sequences were identified in METTL9-interacting proteins, human proteins homologous to ANGST HxH-containing mouse proteins from the ProSeAM experiments, and putative methylhistidine-containing peptides identified through exploratory and further analysis of previously published HeLa proteomics datasets[28]. In addition, peptides were derived from sequences reported as histidine-methylated in literature, various zinc transporters and other HxH-containing proteins of particular biological interest. Peptide sequence windows were chosen to exclude Cys and Trp residues prone to non-enzyme-catalyzed methylation. As negative controls, all HxH-sequences in WT peptides were disrupted by appropriate His-to-Ala mutations. The negative controls were spotted next to the corresponding HxH-containing sequences. The quantitative analysis of array methylation data was performed using ImageJ[46].

**Mitochondrial respiration analyses.** HAP1 cells were seeded in three T75 flasks (8 × 10^6 cells/flask) and grown for 48 h at 37 °C. Cells of three flasks were combined after washing the cells with PBS. Mitochondria from WT and *METTL9* KO cells were isolated as previously described[47]. In brief, pelleted HAP1 cells were resuspended in MIB buffer (210 mM D-mannitol, 70 mM sucrose, 5 mM HEPES, 1 mM EGTA, and 0.5% (w/v) fatty acid-free BSA, pH 7.2), transferred into a glass tube, and disrupted by 30 strokes with a homogenizer. After centrifugation (600×g, 10 min, 4 °C), the supernatant was collected into a new tube and centrifuged at 8000 × g (10 min 4 °C). The pellet was washed twice with MIB buffer; the pellet, containing the mitochondria, was then resuspended in MAS buffer (220 mM d-Mannitol, 70 mM sucrose, 10 mM KH_2PO_4, 5 mM MgCl_2, 2 mM HEPES, 1 mM EGTA, and 0.2% (w/v) of fatty acid-free BSA, pH 7.2) supplemented with 10 mM succinate and 2 µM rotenone or 5 mM malate and 10 mM glutamate, to measure Complex II and I respiration, respectively. Mitochondria (15 ug for Complex I- and 5 ug for Complex II-driven respiration) were added in a non-coated XF24 plates and centrifuged. Oxygen consumption rates (OCRs) were measured under basal conditions, and after sequential addition of ADP (2 mM), oligomycin (3.2 µM), FCCP (4 µM), and antimycin A (4 µM). Each assay cycle consisted of 1 min of mixing and 3 min of OCR measurements. For each condition, three cycles were used to determine the average OCR under a given condition.

Mitochondrial respiration analysis in permeabilized HEK293T cells was performed by following an established method[48] with some modifications. HEK293T cells (2 × 10^4 cells/well in 80 µL DMEM) seeded in microplates were incubated overnight at 37 °C under 5% CO_2. The culture medium was replaced with MAS buffer (220 mM mannitol, 70 mM sucrose, 10 mM KH_2PO_4, 5 mM MgCl_2, 2 mM HEPES-KOH, and 1 mM ethylene glycol tetra acetic acid, pH 7.2), after which the cells were incubated for 15 min at 37 °C in a CO_2-free incubator. Next, three baseline measurements were taken using the Seahorse XFe96 analyzer, after which 25 µL of MAS buffer containing either digitonin (25 µg/mL, final concentration), 1 mM ADP, 1 mM sodium malate and 10 mM sodium pyruvate (for the assessment of Complex I) or digitonin (25 µg/mL), 1 mM ADP, 10 mM

sodium succinate and 0.5 μM rotenone (for Complex II) was injected from port A to start the reaction. This was followed by sequential treatments with oligomycin A (2 μM) from port B and antimycin A (0.5 μM) from port C. Each assay cycle consisted of 2 min of mixing, 2 min of incubation and 3 min of OCR measurements. For each condition, at least three replications were used to determine the average OCR.

**ProSeAM assay**. ProSeAM were prepared as reported in[49]. ProSeAM substrate screening was carried out as reported[50] with some modifications. *Mettl9* KO MEF cells were isolated from E13.5 mouse embryos, cultured in DMEM containing either light Arg and Lys or heavy isotope labeled Arg ($^{13}C_6$ $^{15}N_4$ L-Arginine) and Lys ($^{13}C_6$ $^{15}N_2$ L-Lysine, Thermo Scientific) for at least six doubling times. The cells were harvested and lysed in a lysis buffer (50 mM Tris-HCl with pH 8.0, 50 mM KCl, 10% Glycerol, 1% *n*-Dodecyl-β-D-maltoside). The cell lysates containing 150 μg of proteins were incubated with 150 μM of ProSeAM with (Heavy) or without (Light) 10 μg of mMETTL9 (protein:enzyme = 15:1) in MTase reaction buffer (50 mM Tris-HCl, pH 8.0) at 20 °C for 2 h. The reaction was stopped by adding 4 volumes of ice-cold acetone. The reaction tube was centrifuged at 15,000×g for 5 min, and precipitates were washed once with ice-cold acetone. The pellet was dissolved in 58.5 μL of 1× PBS and 0.2% SDS, after which 15 μL of 5× click reaction buffer and 1.5 μL of 10 mM Azide-PEG4-Biotin (Click Chemistry Tools) were added; the reaction mixture was incubated for 60 min at room temperature (RT). The click reaction was stopped with 4 volumes of ice-cold acetone. The pellet was resuspended in 75 μL of binding buffer (1× PBS, 0.1% Tween-20, 2% SDS, 20 mM dithiothreitol (DTT)) and sonicated for 10 s. The Light and Heavy samples were mixed in a tube; 450 μL of IP buffer (TBS, 0.1% Tween-20) containing 3 μg of Dynabeads M-280 Streptavidin (Life Technologies Japan Ltd., Minato-ku, Tokyo, Japan) was added to the tube, and it was incubated for 30 min at RT (the final SDS concentration in the reaction was 0.5%). The protein-bound beads were washed three times using wash buffer (1× PBS, 0.1% Tween-20, 0.5% SDS) and twice using 100 mM ammonium bicarbonate (ABC) buffer, and were analyzed by western blotting or mass spectrometry. For SILAC MS/MS analysis, DTT (20 mM) was added to protein-bound Dynabeads in 100 mM ABC buffer, and the mixture was incubated for 30 min at 56 °C. Then, iodoacetamide was added and the mixture was incubated for 30 min at 37 °C in the dark. The protein samples were then digested with 1 μg trypsin (Promega). The protein fragments were applied to a liquid chromatograph (EASY-nLC 1000; Thermo Fisher Scientific, Odense, Denmark) coupled to a Q Exactive Hybrid Quadrupole-Orbitrap Mass Spectrometer (Thermo Fisher Scientific, Inc., San Jose, CA, USA), with a nanospray ion source in positive mode. The peptides derived from the protein fragments were separated on a NANO-HPLC C18 capillary column (0.075-mm inner diameter × 150 mm length, 3 mm particle size; Nikkyo Technos, Tokyo, Japan). Mobile phase A was comprised of water with 0.1% formic acid, and mobile phase B was comprised of acetonitrile with 0.1% formic acid. Two different slopes were used for a gradient elution for 120 min at a flow rate of 300 nL/min: 0–30% of phase B in 100 min and 30–65% of phase B in 20 min. The mass spectrometer was operated in the top-10 data-dependent scan mode. The parameters for operating the mass spectrometer were as follows: spray voltage, 2.3 kV; capillary temperature, 275 °C; mass-to-charge ratio, 350–1800; normalized collision energy, 28%. Raw data were acquired using the Xcalibur ver. 4.0 software (Thermo Fisher Scientific). The MS and MS/MS data were searched against the Swiss-Prot database using Proteome Discoverer 1.4 (Thermo Fisher Scientific) using the MASCOT search engine software version no. 2.6.0 (Matrix Science, London, United Kingdom). The peptides were considered identified when their false discovery rates (FDR) were less than 1%. For substrate identification, proteins exhibiting at least a 2-fold increase with at least 5% coverage were defined as positive hit proteins.

**Isothermal titration calorimetry**. SLC39A7 peptides that contain either no (Me0; HGHSHAHGHGHTH) or six 1MH (Me6, HGXSXAXGXGXTX; X = 1MH)) residues were synthesized with a peptide synthesizer (ABI, 433 A) using Fmoc-3-methyl-L-histidine (Merck) for the 1MH residue. The peptides were purified with HPLC, and purity determined by MALDI-MS. Dry weights were quantified with an amino acid analyzer (Hitachi, L-8900) for the calculation of molarity. ITC assay was performed as reported previously[51], with a slight modification. Briefly, SLC39A7 peptides and $Zn^{2+}$ solutions were prepared in 20 mM Tris –HCl pH 7.5, 100 mM NaCl. Titration of $ZnCl_2$ (2 mM) into peptide solutions (50 μM) was performed at 30 °C using an ITC titration calorimeter (Malvern Panalytical, MicroCal iTC200). 39 injections of 1 μL were made with an equilibration time of 1 min between injections. Integration of the thermograms after correction for heats of dilution yielded binding isotherms that fit best to a one-set of sites binding model using the ITC data analysis software Origin 7.0 (MicroCal Inc., Piscataway, NJ). A non-linear least-squares curve-fitting algorithm was used to determine the stoichiometric ratio ($n$), the dissociation constant ($K_d$), and the change in enthalpy ($\Delta H$) of the interaction. All ITC experiments were performed in triplicate.

**Sequence alignments and structural modeling**. Multiple protein sequence alignments were performed using the Muscle algorithm embedded in Jalview[52]. The structure of hMETTL9 was modeled using RaptorX[53] using the Ado-Met-dependent methyltransferase Q8PUk2 from *Methanosarcina mazie* as template (PDB ID 3sm3).

**Fluorescence microscopy**. HeLa cells were transfected with pEGFP-N1 hMETTL9-GFP constructs using FuGENE transfection reagent (Promega) for 24 h and probed with ER-tracker and Mitotracker (ThermoFisher Scientific). Live cells were imaged with PlanApo ×100, numerical aperture 1.1 oil objective (Olympus).

**Reporting summary**. Further information on research design is available in the Nature Research Reporting Summary linked to this article.

## Data availability

The immunoprecipitation mass spectrometry proteomics data have been deposited to the ProteomeXchange Consortium via the PRIDE[54] partner repository with the dataset identifier PXD016408. The DNAJB12 methylation data have been deposited to ProteomeXchange, ID: PXD020010. The ProSeAm SILAC screen data have been deposited at ProteomeXchange, ID: PXD016823. NDUFB3 and S100A9 methylation data are available via ProteomeXchange with identifier PXD022067. The following publicly available databases were used: UniProt [https://www.uniprot.org/] for obtaining protein sequences, and RCSB Protein Data Bank (RCSB PDB) [https://www.rcsb.org/] for obtaining protein structures.
Source data are provided with this paper.

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

## Acknowledgements

The authors thank Bernd Thiede and Per Magne Ueland for useful discussions. We thank M. Shirouzu for help with recombinant protein production and M. Ledsaak, C. Andreassen, and I.F. Kjønstad for help in construct preparation. We thank the staff of Research Resources Division, RIKEN Center for Brain Science, for generation of *Mettl9* KO mice, peptide synthesis, and LC-MS/MS analysis and especially Masaya Usui and Hiromasa Morishita for MRM Quantitation; T. Yamazaki and T. Hirose for providing the hCas9 and dual gRNA plasmid, PX330-B/B, M. Ikeda (RIKEN BDR) for sample preparation and Y. Takeda and Y. Araki for the kind support of mouse neutrophil preparation. We thank Dr. Boudewijn Burgering from University Medical Center Utrecht for the use of Seahorse analyzer instrument. We thank Oslo NorMIC Imaging Platform (Department of Biosciences, University of Oslo) for the use of cell imaging equipment. We thank the Proteoforms@LU proteomics platform at Lund University for instrument support and assistance. This work was supported by the Research Council of Norway (to P.Ø.F.), the Norwegian Cancer Society (to P.Ø.F), the 'Epigenome Manipulation Project' of the All-RIKEN Projects (to Y.Sh., M.So., T.S., Y. So. T.U.); the Japan Ministry of Education, Culture, Sports, Science, and Technology Grant-in-Aid for Scientific Research (16K18476) (to T.Sh.); the Lundbeck Foundation [R231-2016-2682 to M. E.J.]; Novo Nordisk Foundation [NNF16OC0022946 to M.E.J., NNF14CC0001 to J.V. O.]; the Crafoord Foundation (to M.E.J.). This work has been supported by the DFG grant JE 252/7-4 (to A.J.). M.A.M and C.J.S. thank the Wellcome Trust and Cancer Research UK for funding.

## Author contributions

E.D. and T.Sh. designed and performed experiments, analyzed and presented data and prepared the manuscript. M.Sc. and S.W. performed and interpreted peptide array experiments. M.E.J. designed and performed MS and bioinformatics analyses of the GFP-METTL9 immunoprecipitation experiment and M.E.J. and V.S. performed proteomics analysis of METTL9 KO and WT cells. T.L. performed amino acid analysis. H.L.D.M.W. performed Seahorse experiment and analyses. A.Y.Y.H. performed cellular imaging experiments. R.P., J.M., and L.S. performed biochemical experiments. T. Su., N.D., and A.M. performed MS analysis. I.A.G. performed DNA cloning. Y. So., M.A. and M. So. provided ProSeAM. M.K. and T.U. prepared recombinant His-mMETTL9 from insect cells. J.V.O., M.A.M., N.E., and C.J.S. planned and analyzed experiments. A.J. planned experiments and analyzed and presented data. Y.Sh. and P.Ø.F. supervised the study, planned experiments, interpreted data, and prepared the manuscript.

## Competing interests

The authors declare no competing interests.
