## [Peer Review File · Nature Communications]

REVIEWER COMMENTS

Reviewer #1 (Remarks to the Author):

The revised manuscript has addressed most of the issues, particularly the lack of mass spectrometric analysis in the previous version. Still data and presentation of protein mass spectrometry should be more detailed. For all peaks of mass spectrometry, the theoretical m/z value, observed value, difference and ppm should be provided to assess data quality and certainty of peak assignment. High resolution mass spectrometry should also be performed and reported. Moreover, for LC-MS, the retention time or extracted ion chromatography (XIC) should be provided for relevant species.

Reviewer #2 (Remarks to the Author):

I read carefully with the authors' responses to my previous critiques. Below are a few additional comments mainly on the significance of the study as well as clarity of the manuscript.

1. As the authors stated in the reply-to-reviewers, and in the Discussion, that "METTL9 knock-out yielded the spectacular "molecular phenotype" of dramatically reduced 1MH levels across the entire proteome (as measured by amino acid analysis)", it should be at least mentioned whether there is any noticeable phenotype of KO mice in light of such dramatic reduction of 1MH level. Are the Mettl9 KO (-/-) mice normal?
2. The effect on metal Zn binding by the 13-mer peptide (Figure 5c) is not significant. First, the binding affinity of 5 μ M is very low for known metal binding proteins under physiological conditions. Second, the 2-fold change in the μ M range (from 5 to 10 μ M) is neither "significant" (as in the Discussion) nor "diminishes zinc binding" (as stated in Abstract and Introduction). Third, the methylation may not occur to all 6 His residues (it could vary between 1-to-5 as shown in Figure 1g). Does the degree of methylation (1-5 me) has the same effect on Zn binding? It was surprising that

the authors choose to highlight this piece of insignificant in vitro data, while not mentioning any insightful findings from the KO mice. This concerns me regarding the significance of the 1MH methylation (and the study itself)!

As revealed in the current manuscript, His6-tagged METTL9 has extensive automethylation. Does such automethylation diminish the His6-METTL9 binding of Ni(II) ion and thus affect yield of protein purification? Although this is an artificial situation, it should be a better example than the 13-mer peptide, particularly some substrate proteins contain N-terminal HxH motifs.

3. It is unsatisfactory that, as a central substrate protein in the study, NDUFB3 is the only example where methylation was observed in the context of the recombinant protein, but not in the corresponding peptide sequence, whereas the sequence fits all the requirements established in the paper. What “other sequence/structural features” in the context of a short peptide that might have prevented methylation? Are there any common features among the six peptides that failed to be methylated? I can understand the opposite situation where a peptide is methylated but not in the context of folded protein.

4. I understand the data were from three laboratories and different reagents were used in different experiments. Thus it is extremely important to be clear what was used in each experiment and the labels in the figures should be consistent throughout. Human and mouse enzymes should be distinguished as hMETTL9 and mMettl9 as well as tags (GST, His6, N- or C-terminal tag, etc) if not cleaved (for example GST-METTL9 or His6-Mettl9, etc). The same should apply to the substrate proteins used in the study (particularly knowing His6 tag is an in vitro substrate) – please label them in the figures.

Page 2: last line, ref 7 should be replaced by ref 6

Page 4: line 104 recombinant GST-METTL9. Please label the same in the corresponding figures: such as in Figure 1a and supplemental Figure S1a, change METTL9 to GST-METTL9

Page 5: line 123, recombinant proteins – are these tagged proteins expressed in *E. coli*? If so, please state clearly and label them clearly in Fig. 1b and 1c

Page 7 (line 182); page 8 (line 197): recombinant METTL9, which one (GST- or His6-tagged)?

Page 9 (line 224): METTL9 in the presence of ProSeAM – which METTL9 was used? (from p18, it was described as mMETTL9 expressed in baculovirus-infected Sf9 insect cells

Page 10, line 240, Figure 3C shows 50 proteins (56 peptides) (the number is different from Table S1).

Page 10, line 259, E. coli expressed recombinant protein – please specify which tag was used in the expression since Hig6 tag itself is a potential substrate. In Figure 4a, please specify tag for both substrate proteins and METTL9 enzyme used in the assay.

Page 11, lines 279-280: are these Mettl9 KO cells and WT cells or KO and WT mouse neutrophil tissue fractions?

Figure 4b and 4c, please label which protein and source (i.e. mouse or HEK293 expression)

Line 296, METTL9 KO – if it is mouse, it should be in lowercase for Mettl9 KO

Nine METTL9 KO cells/tissues (how many mouse tissues and how many cell lines)?

P18, line 437, recombinant GST- or His6-tagged proteins – please list all constructs of proteins used.

P33, line 797, PDB ID 3sm3A should be 3sm3.

Reviewer #3 (Remarks to the Author):

The authors have satisfactorily addressed my comments and concerns in the revised manuscript.

Point-by-point response to reviewers' comments

(Davydova et al. - NCOMMS-20-07648A)

("The methyltransferase METTL9 mediates pervasive 1- methylhistidine modification in mammalian proteomes")

We thank the reviewers for their critical reading of our revised manuscript and for their useful comments, and were content to see that the revision of our manuscript had alleviated their main concerns.

In addition to the changes made in response to the reviewers, we have also made a few (not very important) corrections to Table 1, i.e. removed one peptide substrate that was not tested on the peptide array, and two peptides substrates where the activity was very low, and, we now realize, slightly below the set threshold (background level).

Our response (normal font) to the comments (*italics*) from reviewers #1 and #2 is found below. The full reviewers' comments are found at the end of this document.

Response to reviewer #1

(broken down into two comments, #1 and #2)

The revised manuscript has addressed most of the issues, particularly the lack of mass spectrometric analysis in the previous version. Still data and presentation of protein mass spectrometry should be more detailed. For all peaks of mass spectrometry, the theoretical m/z value, observed value, difference and ppm should be provided to assess data quality and certainly of peak assignment. High resolution mass spectrometry should also be performed and reported. Moreover, for LC-MS, the retention time or extracted ion chromatography (XIC) should be provided for relevant species.

Comment #1

High resolution mass spectrometry should also be performed and reported.

Our response

We are uncertain on how to potentially respond to this rather unspecific request. First, we should point out that both high resolution mass spectrometry (Orbitrap MS) and mass spectrometry of somewhat lower resolution, i.e. MALDI MS, was used to generate the protein MS data in the present study. However, as already alluded to in our revised manuscript, the His-rich METTL9 substrate sequences are generally difficult to study by LC-MS, and we therefore resorted to MALDI MS to study in vivo methylation of some of the key METTL9 substrates. Thus, attempting to redo these experiments using high resolution MS would represent a very large and time-consuming task with limited chance of success.

Changes to the manuscript

No changes were introduced based on this comment.

Comment #2

Still data and presentation of protein mass spectrometry should be more detailed. For all peaks of mass spectrometry, the theoretical m/z value, observed value, difference and ppm should be provided to assess data quality and certainly of peak assignment. Moreover, for LC-MS, the retention time or extracted ion chromatography (XIC) should be provided for relevant species.

Our response and changes to the manuscript

Based on Comment #1 (*High resolution mass spectrometry should also be performed and reported.*), we have interpreted that the request for specific values (*theoretical m/z value, observed value, difference and ppm*) primarily relates to the MALDI MS experiments. The MALDI MS is of somewhat lower resolution, and such values are therefore particularly important and relevant. However, we have also, in some cases, provided additional information also for the high resolution MS experiments.

For the MALDI MS experiments (shown in Fig. 4b and Fig. 4c), we have now made a table (shown as Supplementary Data 1), where theoretical and experimental m/z (or mass) values, as well as ppm deviation, are shown for all the relevant peaks. To support the MALDI MS results, we have also included MASCOT search results from these experiments (shown as Supplementary Data 2). These results (peptide mass fingerprinting) further confirm the identity of the investigated proteins, and also give information on the experimental and theoretical mass values for the peaks corresponding to the methylated peptides. For the MALDI MS experiments, further additional data/information (which was also present in the previous version of our manuscript) is also included, i.e. MS/MS fragmentation spectra for the methylated peptides (Supplementary Fig. 9), and corresponding fragment ion mass tables (included as Source Data). Finally, the MALDI MS data has now (since our previous manuscript submission) been uploaded to the PRIDE/ ProteomeXchange repository, where also all our other MS data on in vivo methylation can be found.

Regarding "*Moreover, for LC-MS, the retention time or extracted ion chromatography (XIC) should be provided for relevant species.*", we agree that the compact presentation of (high resolution) LC-MS in Fig. 1b and Fig. 1g offers little detail. Therefore, for these experiments, the full XICs, including instrument settings, retention times and calculated peak areas, have been included as Supplementary Figures 3 and 6, respectively.

Response to reviewer #2

Comment #1

As the authors stated in the reply-to-reviewers, and in the Discussion, that "METTL9 knock-out yielded the spectacular "molecular phenotype" of dramatically reduced 1MH levels across the entire proteome (as measured by amino acid analysis)", it should be at least mentioned whether there is any noticeable phenotype of KO mice in light of such dramatic reduction of 1MH level. Are the Mettl9 KO (-/-) mice normal?

Our response

The Mettl9 KO mice are indeed without any overt phenotype, as was already indicated in the Discussion section of the previous version of our manuscript: "... the Mettl9 KO mice, which are without overt phenotype" (P16, line 403). However, we agree that this contrasts with the strong molecular phenotype, and merits some discussion (as also suggested by the editor).

Changes to the manuscript

We have now added a paragraph to the Discussion section, elaborating how the spectacular "molecular phenotype" of dramatically reduced 1MH levels in the Mettl9 KO mice can be reconciled with the lack of overt phenotype.

Comment #2

(further broken down into "#2a", "#2b", "#2c" and "#2d", as well as "#2 - overall").

The effect on metal Zn binding by the 13-mer peptide (Figure 5c) is not significant. First, the binding affinity of 5 μ M is very low for known metal binding proteins under physiological conditions. Second,

the 2-fold change in the μM range (from 5 to 10 μM) is neither “significant” (as in the Discussion) nor “diminishes zinc binding” (as stated in Abstract and Introduction). Third, the methylation may not occur to all 6 His residues (it could vary between 1-to-5 as shown in Figure 1g). Does the degree of methylation (1-5 me) has the same effect on Zn binding? It was surprising that the authors choose to highlight this piece of insignificant in vitro data, while not mentioning any insightful findings from the KO mice. This concerns me regarding the significance of the 1MH methylation (and the study itself)! As revealed in the current manuscript, His6-tagged METTL9 has extensive automethylation. Does such automethylation diminish the His6-METTL9 binding of Ni(II) ion and thus affect yield of protein purification? Although this is an artificial situation, it should be a better example than the 13-mer peptide, particularly some substrate proteins contain N-terminal HxH motifs.

#2a

"First, the binding affinity of 5 μM is very low for known metal binding proteins under physiological conditions."

Our response

Clearly, the intracellular (i.e. cytosolic) concentration of "free" zinc, the relevant metal ion, is very low, i.e. in the picomolar range. Accordingly, many zinc binding-proteins/domains, e.g. zinc fingers, bind zinc with very high affinity (nanomolar to femtomolar). However, the total cellular concentration of zinc (including organellar and complexed zinc) is much higher, i.e. 200 - 300 μM , and cells typically require 0.1 - 1 mM (extracellular) zinc for growth (Liang et al., PMID 27010344; Eide, PMID 32192376)

In agreement with the above, micromolar zinc affinities have indeed been reported for cellular zinc transporters (e.g. Zhang, PMID 31164399), and our test sequence was derived from such a protein (SLC39A/ZIP9). Also, zinc transporters typically contain several histidine-containing, zinc binding motifs/sequences that have been shown to possess both low affinity (micromolar) and high affinity (nanomolar) binding sites for zinc (Antala and Dempski, PMID 22242765). Moreover, several articles have shown that stretches/loops of alternating histidines, i.e. sequences similar to our test sequence, play important functional roles in the zinc transporters. Actually, the zinc affinity of one such sequence was investigated by isothermal titration calorimetry (the same method as we used), and found to be in the micromolar range (Tanaka et al., PMID 23772397).

Based on the above, we argue that the measured binding affinity is within a biologically relevant range (especially, in the context of zinc transporters).

#2b

"Second, the 2-fold change in the μM range (from 5 to 10 μM) is neither “significant” (as in the Discussion) nor “diminishes zinc binding” (as stated in Abstract and Introduction)."

Our response

When using the term "significant", we have primarily used it in the sense "statistically significant", which is clearly the case for our data. In this regard, it may be mentioned that, in the zinc binding experiments for the first version of our paper, we experienced some technical problems, and the data showed a rather large spread (but still showed a statistically significant difference between methylated and unmethylated peptide). The relatively large scatter of these data were commented upon by the reviewer (#2), and we did, for the revised manuscript, generate a new set of highly improved data, with a much higher (statistical) significance level.

Given the complexity of the regulation of cellular zinc levels, it is in our view likely that a two-fold shift in affinity may indeed have the potential to regulate cellular zinc biology. However, we do agree that the potential biological/functional significance of these data was somewhat over-emphasized in the previous version of our manuscript; see also below.

#2c

"Third, the methylation may not occur to all 6 His residues (it could vary between 1-to-5 as shown in Figure 1g). Does the degree of methylation (1-5 me) has the same effect on Zn binding? It was surprising that the authors choose to highlight this piece of insignificant in vitro data, while not mentioning any insightful findings from the KO mice. This concerns me regarding the significance of the 1MH methylation (and the study itself)!"

Our response

This comment appears to result from a misunderstanding. The methylated peptide sequence used in this experiment was (unlike in Fig. 1g) not generated by treatment with METTL9 (which would yield a variety of methylation states). Instead, a peptide where six out of the seven histidines were fully modified with 1MH was generated by peptide synthesis; this information could be retrieved from in the figure legend and in the Methods section, but we now see that this may have been pointed out more clearly. In principle, additional peptides with lower degrees of methylation could be synthesized and tested for zinc binding, but we believe that such experiments would not add significantly to the manuscript, since the difference between the fully (on all potential METTL9 sites) and non-methylated peptide is already quite modest. Moreover, as 1MH is a non-standard post-translational modification, the relevant peptides are very expensive (thousands of USD) and have delivery times of several months, making it questionable whether the potential value of these experiments would justify the resources and time spent. Regarding the comment on "*the significance of the 1MH methylation*", this has already been addressed in our response to Comment #1.

#2d

"As revealed in the current manuscript, His6-tagged METTL9 has extensive automethylation. Does such automethylation diminish the His6-METTL9 binding of Ni(II) ion and thus affect yield of protein purification? Although this is an artificial situation, it should be a better example than the 13-mer peptide, particularly some substrate proteins contain N-terminal HxH motifs."

Our response

In the highly sensitive fluorography experiments, where cell or tissue extracts (that contain very small amounts of the individual substrates), were incubated with a large amount of recombinant enzyme, automethylation of His6-tagged METTL9 gave a relatively strong signal, which may potentially mask other substrates. We therefore used GST-METTL9 for such experiments. However, such automethylation is not extensive; we were never able to observe by MS more than ca 10 % monomethylation of (a single histidine in) polyhistidine stretches (and negligible levels of higher methylation states). In agreement with this, there were no indications that automethylation affects the purification of recombinant His6-METTL9, as the purification yields of wild-type His6-METTL9 and the enzyme-dead mutant were similar.

The reviewer suggests that a 6xHis-tag fused to METTL9 may be a better model (than the peptide currently used) to demonstrate how methylation affects binding of zinc to stretches of histidines. However, we disagree on this. First, and as elaborated above, METTL9-mediated methylation of the 6xHis-tag is highly inefficient, in agreement with x = His in the HxH motif being a suboptimal (non-ANGST) residue. Second, none of the HxH-motifs shown to be methylated in vivo have His at the "x" position, indicating that polyhistidine stretches are not very relevant METTL9 substrates. Third, we have actually, for the previous revision, demonstrated that SLC39A7 is subject to METTL9-dependent 1MH modification in vivo, thereby increasing the relevance of the 13-mer SLC39A7-derived model peptide.

#2 - overall

The effect on metal Zn binding by the 13-mer peptide (Figure 5c) is not significant.

Our response

We have demonstrated that methylation leads to a relatively small (2-fold), but statistically significant, reduction in the zinc binding affinity of a relevant model peptide. In our view (as elaborated above), this result may well reflect a mechanism for regulation of cellular metal homeostasis, and therefore should be reported. At least there is nothing in the data that excludes such a possibility, given the complexity and reported micromolar affinities of zinc transporters.

However, we do clearly acknowledge that this is a rather limited finding that will have to be followed up by more rigorous experiments, e.g. testing different divalent metal ions (in addition to zinc) on several different peptides with varying degrees of methylation. Also, we see that that we may have excessively highlighted this finding and its significance in the previous version of our manuscript.

Changes to the manuscript

We have toned down the significance of the zinc binding results throughout the manuscript. Also, it has now been specified in the legend to Fig. 5c that a synthetic peptide was used.

Comment #3

"It is unsatisfactory that, as a central substrate protein in the study, NDUFB3 is the only example where methylation was observed in the context of the recombinant protein, but not in the corresponding peptide sequence, whereas the sequence fits all the requirements established in the paper. What "other sequence/structural features" in the context of a short peptide that might have prevented methylation? Are there any common features among the six peptides that failed to be methylated? I can understand the opposite situation where a peptide is methylated but not in the context of folded protein."

Our response

We agree that the results obtained with NDUFB3 are somewhat counterintuitive, in that the full-length protein was subject to METTL9-dependent methylation in vitro and in vivo, whereas no in vitro methylation of the corresponding peptide was detected. However, our data on histidine methylation of NDUFB3 and its installment by METTL9 are in our view very solid; the "outlier" result is really the lack of methylation in the context of a peptide. Since the methylation sites in NDUFB3 are localized very close to the N-terminal end, we were concerned that the N-terminus of the peptide may somehow play a role (although this is not suggested by our findings on METTL9 substrate specificity). Therefore, we tested peptide versions both with or without the initiator methionine (corresponding to amino acids 1-15 or 2-16, respectively, of the annotated NDUFB3 sequence; UniProt O43676). However, in neither case was methylation observed, supporting that this is a "true" negative result.

The referee asks *"What "other sequence/structural features" in the context of a short peptide that might have prevented methylation?"* but we believe the question may rather be turned around, i. e. into *"What are the structural features that may promote methylation of the NDUFB3 sequence in the context of full-length NDUFB3"* (but that are absent in the peptide). Our data demonstrate that the presence of an ANGST HxH motif is not always sufficient for methylation to occur (see also below); thus, it is clearly conceivable that a structuring of the relevant NDUFB3 sequence, e.g. through interaction with other parts of the protein, somehow promotes methylation in the context of the full-length protein.

Regarding the question *"Are there any common features among the six peptides that failed to be methylated?"* it is not so surprising that some peptides failed to be methylated given that our data clearly demonstrate that an ANGST HxH motif is necessary, but not always sufficient, for METTL9-mediated methylation to occur. This is best illustrated by our observation that substitution of residues flanking the HxH motif in several cases abolished methylation, (Supplementary Fig. 5), and that the effect varied between three different test sequences. Thus, in addition to certain residues clearly being disfavored at the flanking positions, METTL9-mediated methylation of ANGST HxH motifs also appears to be influenced by local sequence context in a somewhat unpredictable manner. Thus, it is, in our view, not at all surprising that, among a set of ANGST HxH-containing METTL9 candidate peptide substrates, some are not subject to methylation. It may be also mentioned that some, but not all, of the ANGST HxH peptides that failed to be methylated, contained an incompatible flanking residue.[Note: We have

actually discovered that one of the "six peptides" wrongly had been annotated as an ANGST HxH peptide; it should have been annotated as a non-ANGST HxH peptide. This has been corrected in the revised version of the manuscript.]

In summary, the referee has brought forward some important points that may have been inadequately discussed and explained in the previous version of our manuscript.

Changes to the manuscript

We have now rewritten and expanded the relevant part of the Discussion section, paying particular attention to the issues brought up by the referee.

Comment #4

"I understand the data were from three laboratories and different reagents were used in different experiments. Thus it is extremely important to be clear what was used in each experiment and the labels in the figures should be consistent throughout. Human and mouse enzymes should be distinguished as hMETTL9 and mMETTL9 as well as tags (GST, His6, N- or C-terminal tag, etc) if not cleaved (for example GST-METTL9 or His6-METTL9, etc). The same should apply to the substrate proteins used in the study (particularly knowing His6 tag is an in vitro substrate) – please label them in the figures."

Our response

We agree that information on which reagents that were used in the various experiments should be stated more clearly.

Changes to the manuscript

The requested information has been added to the manuscript text and to the figure legends.

Non-numbered comments

Note: In many of these comments, the reviewer warrants information on reagents (e.g. on recombinant protein tags). We would like to point out that such information is generally found in the Methods section. However, we agree that it is valuable to provide such information also in Results and Figures, and have complied with the suggestions from the reviewer.

Page 2: last line, ref 7 should be replaced by ref 6

This mistake has been corrected.

Page 4: line 104 recombinant GST-METTL9. Please label the same in the corresponding figures: such as in Figure 1a and supplemental Figure S1a, change METTL9 to GST-METTL9

The suggested changes have been introduced.

Page 5: line 123, recombinant proteins – are these tagged proteins expressed in E. coli? If so, please state clearly and label them clearly in Fig. 1b and 1c

Yes, these are His6-tagged proteins expressed in E. coli. The indicated information has been added.

Page 7 (line 182); page 8 (line 197): recombinant METTL9, which one (GST- or His6-tagged)?

For these purposes, His6-tagged METTL9 was used. This has now been indicated in the text.

Page 9 (line 224): METTL9 in the presence of ProSeAM – which METTL9 was used? (from p18, it was described as mMETTL9 expressed in baculovirus-infected Sf9 insect cells

It is correct that (baculovirus-expressed) mMETTL9 was used for this purpose. This has now been indicated.

Page 10, line 240, Figure 3C shows 50 proteins (56 peptides) (the number is different from Table S1). It is correct that 51 proteins (57 peptides) was included on the array, and are indicated in Table S1, whereas only 56 HxH-containing peptides, derived from 50 proteins, are indicated in the flow chart in Fig. 3c.

This is because the array contained a single HxH-less test peptide, derived from Myosine Light Chain Kinase 2 (MYLK2) which represents one out of two mammalian 1MH-containing proteins reported prior to our study (the other being S100A9). Actually, this was already indicated in the legend to Fig. 3: "Note that the array also contained a single HxH-less peptide, based on the previously reported histidine-methylated sequence from MYLK2.". However, the sentence has now been modified to avoid misunderstanding.

Page 10, line 259, *E. coli* expressed recombinant protein – please specify which tag was used in the expression since Hig6 tag itself is a potential substrate. In Figure 4a, please specify tag for both substrate proteins and METTL9 enzyme used in the assay.

We have now indicated the tags, as requested. However, related to the comment "...His₆ tag itself is a potential substrate" we should mention that one of the substrate proteins used did contain a His₆ tag (His₆-ARMC6). However, this did not cause any experimental problems; as also elaborated above, methylation of the His₆ tag is inefficient and therefore negligible compared to the methylation of the relevant HxH motif(s). This is clearly demonstrated by the observation that mutation of the (internal) HxH motif (while leaving the N-terminal His₆-tag intact) abolished methylation of His₆-ARMC6 (Fig. 1c and 4a).

Page 11, lines 279-280: are these *Mettl9* KO cells and WT cells or KO and WT mouse neutrophil tissue fractions?

Actually, this was stated a few lines above (l. 276): " However, we were able to isolate S100A9 from mouse peritoneal exudate neutrophils.....". This has now, for clarity, also been specified in the relevant sentence, replacing e.g. "KO cells" with "KO neutrophils"

Figure 4b and 4c, please label which protein and source (i.e. mouse or HEK293 expression)

This information was already given in the figure legend, but has now also been indicated directly on the figure.

Line 296, METTL9 KO – if it is mouse, it should be in lowercase for *Mettl9* KO

In this case, it was referred to results both from mouse and human. We therefore chose to use only uppercase, which we think is acceptable. (The alternative METTL9/*Mettl9*, would have appeared a bit odd, in our view)

"Nine METTL9 KO cells/tissues (how many mouse tissues and how many cell lines)?"

(we assume this refers to P14, line 349, in the Discussion section) These were six tissues and one MEF cell line from mice, as well as two human cell lines. This information could be inferred from the relevant figure (Fig. 2c,d), but we realize it is useful to also include it in the text, and this has now been done.

P18, line 437, recombinant GST- or His₆-tagged proteins – please list all constructs of proteins used. This information has now been added.

P33, line 797, PDB ID 3sm3A should be 3sm3.

This has been corrected ("A" was added to indicate that the A chain that was used for modelling, but since this structure only consists of a single chain, the information was redundant).